# Cdc6 is sequentially regulated by PP2A-Cdc55, Cdc14, and Sic1 for origin licensing in *S. cerevisiae*

**Jasmin Philip[1,2], Mihkel Örd[3], Andriele Silva[1,2], Shaneen Singh[1,2], John FX Diffley[4], Dirk Remus[5], Mart Loog[3], Amy E Ikui[1,2]***

[1]The PhD Program in Biochemistry, The Graduate Center, CUNY, Brooklyn, United States; [2]Brooklyn College, Brooklyn, United States; [3]University of Tartu, Tartu, Estonia; [4]The Francis Crick Institute, London, United Kingdom; [5]Memorial Sloan-Kettering Cancer Center, New York, United States

**Abstract** Cdc6, a subunit of the pre-replicative complex (pre-RC), contains multiple regulatory cyclin-dependent kinase (Cdk1) consensus sites, SP or TP motifs. In *Saccharomyces cerevisiae*, Cdk1 phosphorylates Cdc6-T7 to recruit Cks1, the Cdk1 phospho-adaptor in S phase, for subsequent multisite phosphorylation and protein degradation. Cdc6 accumulates in mitosis and is tightly bound by Clb2 through N-terminal phosphorylation in order to prevent premature origin licensing and degradation. It has been extensively studied how Cdc6 phosphorylation is regulated by the cyclin–Cdk1 complex. However, a detailed mechanism on how Cdc6 phosphorylation is reversed by phosphatases has not been elucidated. Here, we show that PP2A$^{Cdc55}$ dephosphorylates Cdc6 N-terminal sites to release Clb2. Cdc14 dephosphorylates the C-terminal phospho-degron, leading to Cdc6 stabilization in mitosis. In addition, Cdk1 inhibitor Sic1 releases Clb2·Cdk1·Cks1 from Cdc6 to load Mcm2–7 on the chromatin upon mitotic exit. Thus, pre-RC assembly and origin licensing are promoted by phosphatases through the attenuation of distinct Cdk1-dependent Cdc6 inhibitory mechanisms.

*For correspondence:
AIkui@brooklyn.cuny.edu

**Competing interest:** The authors declare that no competing interests exist.

## Editor's evaluation

The work at focus in your manuscript shows that there are three mechanisms that coordinate the stability of Cdc6 to allow for the critical DNA replication licensing event that must occur before yeast cells enter the G1 phase and is an important contribution to the cell cycle field. How Cdc6 is dephosphorylated is complex and the reviewers all applaud the biochemical and molecular biology approaches taken. While no new experiments are required for the publication we call your attention to the three reviews below where recommendations for revisions are given.

## Introduction

Pre-replicative complexes (pre-RCs) are assembled on DNA to license replication origins in M–G1 phase. The pre-RC components are recruited in a sequential fashion that starts with the origin recognition complex (Orc1–6) followed by Cdc6 then Cdt1, which eventually load the Mcm2–7 helicase on DNA (*Bell and Stillman, 1992*; *Santocanale and Diffley, 1996*; *Newlon, 1997*; *Labib et al., 2001*; *Tanaka and Diffley, 2002*). At the onset of S phase, cyclin-dependent kinase (Cdk1) phosphorylates pre-RC components such as Cdc6 which prevents reinitiation of DNA replication through multiple mechanisms (*Nguyen et al., 2001*; *Wilmes et al., 2004*). Cdc6 phospho-degrons are created by Cdk1, which directs Cdc6 protein degradation via SCF-mediated ubiquitination in *Saccharomyces*

cerevisiae (**Drury et al., 2000**; **Drury et al., 1997**; **Perkins et al., 2001**). Cdc6 is expressed and stabilized during late mitosis; however, it is not clear how Cdc6 stability is maintained in the presence of high Cdk1 activity.

There are eight Cdk1-dependent phosphorylation sites that contain TP or SP motifs in Cdc6, six of which have been well characterized (**Figure 1A**). Cyclin docking motifs facilitate Cdk1-dependent phosphorylation at specific cell cycle stages (**Wilmes et al., 2004**; **Cross and Jacobson, 2000**; **Loog and Morgan, 2005**; **Kõivomägi et al., 2011**; **Örd et al., 2019**). The S-phase cyclin Clb5 and M-phase cyclin Clb2 associate with the docking motifs RxL and LxF, respectively (**Wilmes et al., 2004**; **Kõivomägi et al., 2011**; **Örd et al., 2019**). Clb5 binds to the Cdc6 RxL motif at amino acid residues 30–32 to mediate phosphorylation of T7 in the Cdc6 N-terminus (**Örd et al., 2019**). The Cdk1 phospho-adaptor Cks1 docks at phosphorylated T7 to trigger a Cdc6 multiphosphorylation cascade in the N- to C-direction (**Örd et al., 2019**). SCF$^{Cdc4}$ recognizes Cdc6 phospho-degrons at T39-S43 and T368-S372 for ubiquitin-mediated degradation under control of the SCF$^{Cdc4}$ ubiquitin ligase (**Figure 1A**; **Drury et al., 2000**; **Drury et al., 1997**; **Perkins et al., 2001**; **Al-Zain et al., 2015**).

Following Cdc6 degradation at G1/S, it is expressed again in mitosis and tightly bound to mitotic cyclin Clb2 to prevent origin licensing in mitosis (**Mimura et al., 2004**). Clb2 interacts with Cdc6 through the Clb2 hydrophobic patch (*hp*), a substrate-binding pocket in cyclins (**Örd et al., 2019**). The specific binding of Clb2 to Cdc6 is mediated through the $^{47}$LQF$^{49}$ motif in Cdc6, which is facilitated by Cks1 docking at phosphorylated T7 in the Cdc6 N-terminus (**Örd et al., 2019**). The Cdc6–Clb2 binding is also enhanced by a $^{126}$FQSLP$^{130}$ motif located in the midregion of Cdc6[14]. Thus, the Clb2·Cdk1·Cks1 complex interacts with Cdc6 through multiple elements: Cks1 docking at phosphorylated T7, and Clb2 binding at the $^{47}$LQF$^{49}$ and $^{126}$FQSLP$^{130}$ motifs (**Örd et al., 2019**). The tight Cdc6–Clb2 interaction potentially shields Cdc6 phospho-degrons, leading to Cdc6 stabilization. These mechanisms could accumulate Cdc6 in mitosis without origin licensing.

Mitotic exit requires a complete suppression of Cdk1 activity to reset the cell cycle for G1 entry. It has been suggested that Cdc6 acts as a Cdk1 inhibitor together with Sic1 and a coactivator of the anaphase-promoting complex (APC), Cdh1 (**Calzada et al., 2001**). Sic1 and Cdc6 share similarities in Cdk1 inhibitory function, as both have been shown to associate to cyclin–Cdk1 complexes with low nanomolar affinity, which is partly facilitated by Cks1 docking (**Örd et al., 2019**; **Venta et al., 2020**). Deletion of the Cdc6 N-terminal Cdk1 site (*cdc6Δ47)* shows elevated Clb2–Cdk1 activity in mitosis, indicating that Cdc6 N-terminus plays a role in Clb2–Cdk1 inhibition (**Calzada et al., 2001**). A combination of *Δsic1* and *cdc6Δ47* leads to defects in mitotic exit, probably due to high Cdk1 activity (**Calzada et al., 2001**). Furthermore, *Δsic1 Δcdh1 cdc6Δ2–49* triple mutants caused cytokinesis defects (**Archambault et al., 2003**). Altogether, Cdc6, Sic1, and APC$^{Cdh1}$ cooperate to inhibit Clb2–Cdk1 activity for mitotic exit. This was further supported by an in vitro assay that shows the inhibition of Clb2–Cdc28 kinase activity by Cdc6 (**Örd et al., 2019**).

Beside these Cdk1 inhibition mechanisms, Cdc6 phosphorylation needs to be removed by phosphatases during late mitosis. It is unknown which phosphatase is responsible for Cdc6 dephosphorylation. It has been reported that Cdc6 physically interacts with Cdc55, a regulatory subunit for protein phosphatase 2A (PP2A) (**Boronat and Campbell, 2007**). PP2A belongs to a family of serine/threonine phosphatases with a well-conserved role in mitosis from yeast to humans (**Janssens and Goris, 2001**; **Mochida et al., 2009**). In *S. cerevisiae*, the PP2A heterotrimeric complex is composed of a scaffold A subunit (Tpd3), a regulatory B subunit (Cdc55, Rts1, or Rts3), and two largely interchangeable catalytic C subunits (Pph21 and Pph22) (**Healy et al., 1991**; **van Zyl et al., 1992**; **Shu et al., 1997**). Among these subunits, the regulatory B subunit primarily dictates cellular localization and PP2A substrate specificity (**Rossio and Yoshida, 2011**). PP2A$^{Cdc55}$ and PP2A$^{Rts1}$ have been implicated in mitotic progression. Cytoplasmic PP2A$^{Cdc55}$ dephosphorylates and inhibits Swe1, a Cdk1 inhibitor, to drive mitotic entry (**Lin and Arndt, 1995**; **Yang et al., 2000**). Nuclear PP2A$^{Cdc55}$ plays a role in the spindle assembly checkpoint and inhibits chromosome segregation by dephosphorylating Cdc20 (**Minshull et al., 1996**; **Wang and Burke, 1997**; **Lianga et al., 2013**; **Rossio et al., 2013**). Furthermore, various stresses inhibit the cell cycle via PP2A$^{Cdc55}$ (**Tang and Wang, 2006**; **Khondker et al., 2020**). PP2A$^{Cdc55}$ also dephosphorylates Net1, an inhibitor of Cdc14 (**Queralt et al., 2006**). PP2A$^{Rts1}$ promotes cytokinesis by modulating actin ring phosphorylation (**Dobbelaere et al., 2003**). It is unclear if PP2A directly targets and dephosphorylates Cdc6.

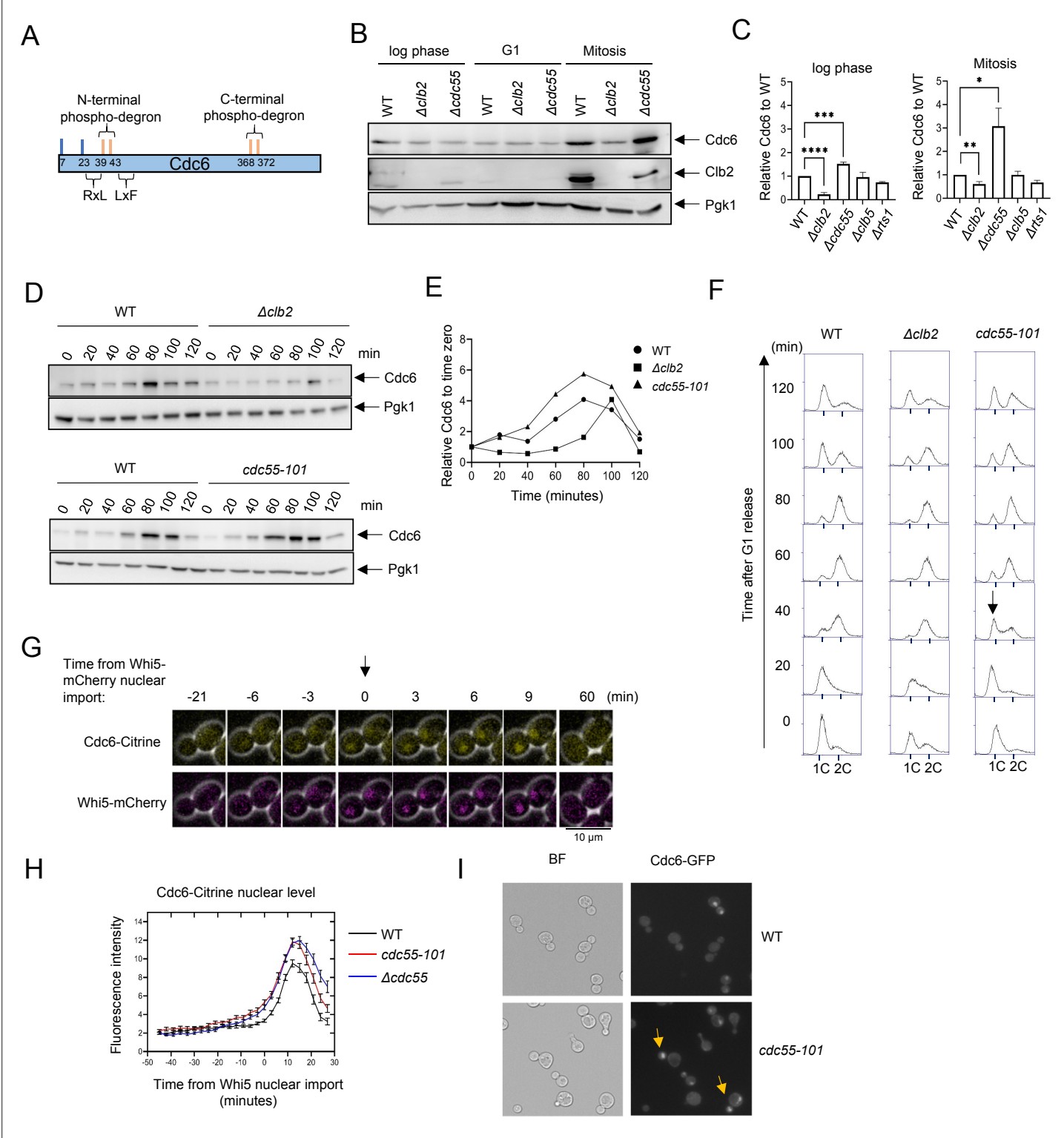

**Figure 1.** Cdc6 is more stabilized in *Δcdc55* cells than in *Δclb2*. (**A**) Cdc6 contains six functional Cdk1 phosphorylation sites including N- and C-terminal phospho-degrons at T39-S43 and S368-T372, respectively (orange bars). RxL and LxF motifs mediate cyclin interaction. (**B**) *CDC6-9MYC* (WT), *CDC6-9MYC Δclb2* (*Δclb2*), or *CDC6-9MYC Δcdc55* (*Δcdc55*) cells were incubated to log phase, or arrested in G1 by α-factor or mitosis by nocodazole. Protein was extracted and subjected to western blot analysis to visualize Cdc6-9MYC and Clb2. Pgk1 was used as a loading control. (**C**) Cdc6 protein levels from A were quantified and normalized to WT. Three independent biological replicates were performed. The average of relative Cdc6 band intensity compared to WT is shown. Error bars are standard error of the mean (SEM; *n* = 3; log phase *Δclb2* ****p < 0.0001, *Δcdc55* ***p = 0.0006, Mitosis *Δclb2*

*Figure 1 continued on next page*

*Figure 1 continued*

\*\*p = 0.0075, *Δcdc55* \*p = 0.0351 was calculated by unpaired Student's *t*-test). (**D**) *CDC6-PrA* (WT), *CDC6-PrA Δclb2* (*Δclb2*), and *CDC6-PrA cdc55-101* (*cdc55-101*) were synchronized in G1 phase by α-factor and released. Samples were collected at indicated times. Cdc6-PrA was visualized by western blotting analysis. Pgk1 was used as a loading control. (**E**) The relative Cdc6 band intensity to time 0 from C is shown. (**F**) Samples from C were fixed and stained with propidium iodide (PI) to show cell cycle profile by flow cytometry. *x*-Axis is PI and *y*-axis is cell number. Arrow indicates S-phase entry delay. (**G**) *CDC6-Citrine WHI5-mCherry* cells were imaged by time-lapse microscopy. Time 0 was set when Whi5-mCherry is imported to the nucleus. Arrow marks Whi5 nuclear entry in late mitosis. (**H**) *CDC6-Citrine* (WT), *CDC6-Citrine cdc55-101* (*cdc55-101*), or *CDC6-Citrine Δcdc55* (*Δcdc55*) with Whi5-mCherry were imaged by time-lapse microscopy. Shown is the average Cdc6-Citrine fluorescence intensities of 100 cells per time point. Error bars are SEM (*n* = 100). (**I**) *CDC6-GFP* (WT) or *CDC6-GFP cdc55-101* (*cdc55-101*) cells were grown to log phase and imaged using a fluorescence microscope.

Cdc14 phosphatase promotes mitotic exit through Net1 dephosphorylation upon Cdc14 release from the nucleolus via the FEAR and MEN networks (*Visintin et al., 1999*; *Azzam et al., 2004*; *Shou et al., 1999*; *Tomson et al., 2009*). Cdc14 dephosphorylates cell cycle regulators such as Swi5, Sic1, and Cdh1, all which are necessary for mitotic exit and G1 entry (*Visintin et al., 1998*; *Jaspersen et al., 1999*). It has been reported that Cdc6 is destabilized in *cdc14-3* mutant cells, which prompted us to study if Cdc14 directly dephosphorylates Cdc6 (*Zhai et al., 2010*). In this study, we show that Cdc6 is directly dephosphorylated by PP2A^Cdc55 and Cdc14 at distinct Cdk1 sites, which releases Clb2 and stabilize Cdc6, respectively. Sic1 also releases Clb2 from Cdc6, allowing Mcm2–7 loading on chromatin for pre-RC assembly. These results suggest a model in which distinct phosphatases relieve Cdk1-dependent Cdc6 inhibition. Finally, a structural insight into Cdc6–Clb2 binding using bioinformatics suggests a role of Cdc6 intrinsically disordered regions (IDRs) for protein–protein interactions.

## Results

### Clb2 and PP2A^Cdc55 have contrasting roles in Cdc6 stability

To test the idea that PP2A^Cdc55 targets Cdc6, we examined Cdc6 protein levels in *Δclb2* and *Δcdc55* cells by western blot analysis. The Cdc6 protein level was lower in *Δclb2* compared to wild type in log-phase cells, which supports previous results of Cdc6 destabilization in the Cdc6-lxf mutant that does not bind to Clb2 (*Figure 1B, C*, *Örd et al., 2019*). There was low Cdc6 protein expression in G1 arrested cells, which was not impacted by deletion of *CLB2* or *CDC55* (*Figure 1B*, middle). During mitosis, we observed less Cdc6 in *Δclb2* cells and more in *Δcdc55* cells compared to wild type; this difference in the Cdc6 protein level was more prominent in mitosis than in log-phase cells (*Figure 1B, C*). These results indicate that PP2A^Cdc55 and Clb2 regulate Cdc6 protein levels in a contradictory manner. An elevated Clb2 protein level and Cdc6 stabilization in wild-type cells confirmed that cells were in mitosis (*Figure 1B*, right). The S-phase cyclin Clb5, as well as another regulatory subunit of PP2A, Rts1, did not alter Cdc6 protein levels (*Figure 1C*). We conclude that Cdc6 is regulated by mitotic cyclin Clb2 and PP2A^Cdc55 in a specific manner.

Next, we analyzed Cdc6 protein levels in *Δclb2* and *cdc55-101* cells during the cell cycle. The *cdc55-101* mutant excludes Cdc55 from the nucleus, allowing us to study Cdc55 nuclear function. Unlike *Δcdc55*, *cdc55-101* mutant cells do not exhibit morphological defects, which allows us to monitor G1 arrest induced by α-factor and cell cycle progression (*Sasaki et al., 2000*). In wild-type cells, Cdc6 protein levels were suppressed throughout S phase, increased in mitosis at 80 min and then diminished (*Figure 1D, E*; *Drury et al., 2000*; *Drury et al., 1997*; *Piatti et al., 1995*). In *Δclb2* cells, Cdc6 protein levels were suppressed throughout S phase and particularly in mitosis (*Figure 1D–F*). This result suggests a role for Clb2 in Cdc6 stabilization during mitosis possibly by masking the Cdc6 phospho-degron. In contrast, Cdc6 was more stabilized in *cdc55-101* cells during mitosis, with a peak accumulation between 60 and 100 min (*Figure 1D, E*). Flow cytometry analysis did not show a significant difference in the cell cycle profile between wild-type and *Δclb2* cells (*Figure 1F*). In *cdc55-101* cells, we observed a 20-min S-phase delay, indicating that PP2A^Cdc55 positively regulates S-phase progression (*Figure 1F*). We conclude that the early Cdc6 accumulation in *cdc55-101* cells is not due to cell cycle delay.

To examine how PP2A^Cdc55 impacts Cdc6 localization and protein level, we analyzed Cdc6-Citrine by time-lapse microscopy in wild-type and *Δcdc55* cells. Whi5 is a Start transcriptional repressor imported into the nucleus in M/G1 and exported in late G1. Whi5 nuclear import was used as a cell cycle indicator that was set as time 0 for a temporal framework (*Figure 1G*; *Doncic et al., 2011*). Cdc6-Citrine

began accumulating in the nucleus after Whi5 nuclear import and reached maximum accumulation at 9 min after Whi5 nuclear import in wild-type cells (***Figure 1H***). In *Δcdc55* and *cdc55-101* cells, we observed more Cdc6-Citrine accumulation (***Figure 1H***). The timing of Cdc6-Citrine expression and degradation remained unchanged, as it started dropping around 15 min after Whi5 nuclear import in all strains (***Figure 1H***). These data suggest that PP2A^Cdc55 negatively regulates Cdc6 stability in the nucleus during mitosis. We also observed Cdc6-GFP nuclear localization in early mitosis in asynchronous *cdc55-101* cells (***Figure 1I***, arrow). The similar Cdc6 protein accumulation patterns observed in *Δcdc55* and *cdc55-101* support the idea that Cdc6 is solely regulated by nuclear PP2A^Cdc55 function.

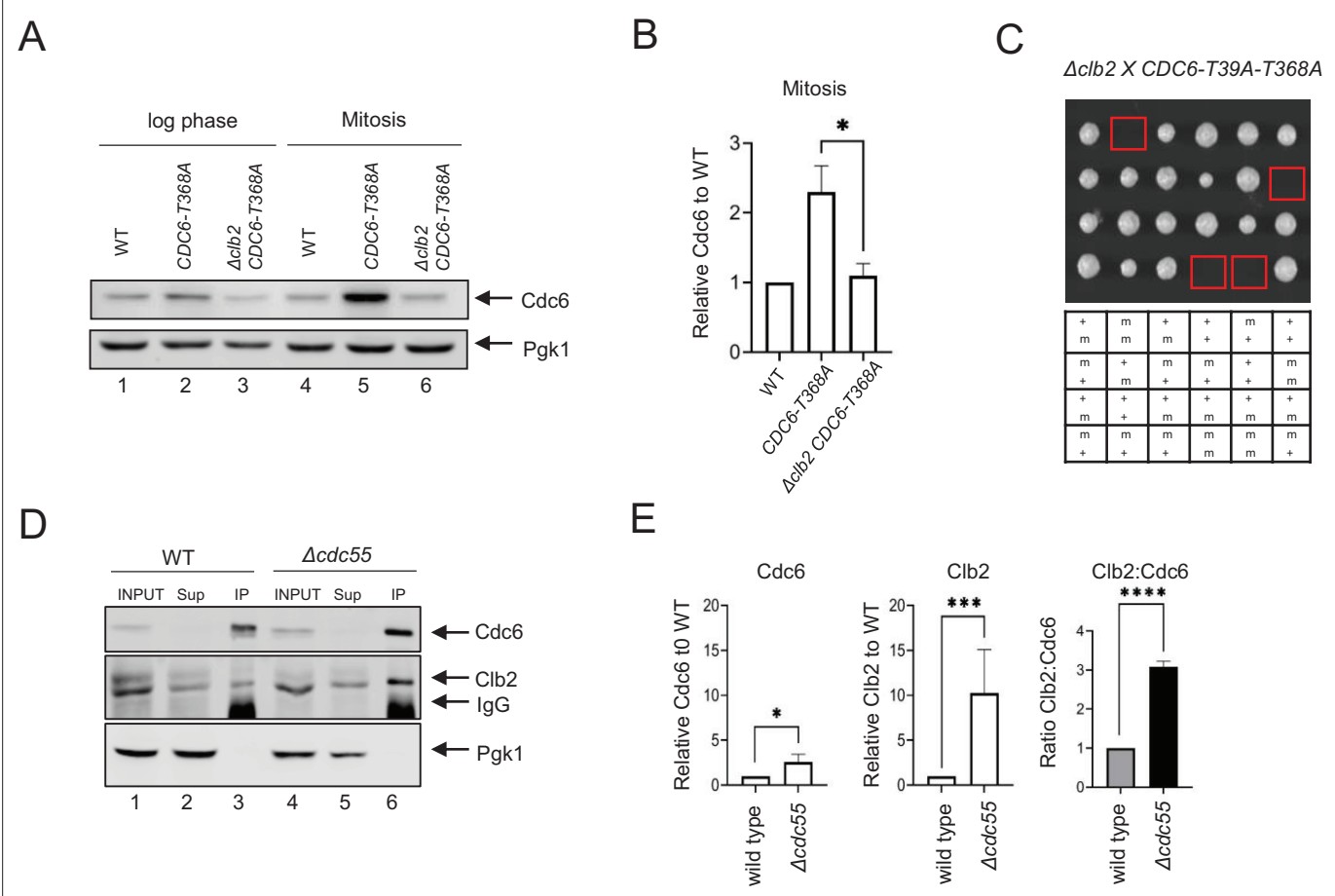

**Figure 2.** Cdc6 and Clb2 are more tightly bound in *Δcdc55* cells. (**A**) *CDC6-PrA* (WT), *CDC6-T368A-PrA, or Δclb2 CDC6-T368A-PrA* cells were incubated to log phase or arrested in mitosis by nocodazole. Protein extracts were subjected to western blot analysis to visualize Cdc6-PrA. Pgk1 was used as a loading control. (**B**) Cdc6 protein levels in A were quantified and normalized to WT. Three independent biological replicates were performed. An average of the relative Cdc6 band intensity is shown. Error bars are standard error of the mean (SEM; *n* = 3; *Δclb2 CDC6-T368A-PrA* *p = 0.0442 was calculated using unpaired Student's *t*-test). (**C**) *Δclb2* strain was crossed with *CDC6-T39A-T368A and* tetrad analysis was performed. The genotype for each haploid progeny is shown. + is wild type and m is mutant. Top letter is for *Δclb2* and bottom letter is for *CDC6-T39A-T368A*. (**D**) *CDC6-9MYC* (WT) or *CDC6-9MYC Δcdc55* (*Δcdc55*) cells were arrested in mitosis by nocodazole. Samples were collected and protein was extracted (INPUT). Cdc6-9MYC was pulled down by anti-MYC agarose beads. Supernatant (Sup) or pull-down samples (IP) were analyzed by western blot to visualize Cdc6-9MYC and Clb2. Pgk1 was used as a loading control. (**E**) Band intensities from E were quantified. Cdc6 and Clb2 levels in *Δcdc55* were normalized to those in WT. Three independent biological replicates of the experiment in E were performed. An average of the relative Cdc6 and Clb2 band intensities are shown. The ratio between Clb2 to Cdc6 was obtained and normalized to WT. Error bars are SEM (*n* = 3; Cdc6 *Δcdc55* *p = 0.0205, Clb2 *Δcdc55* ***p = 0.0003, and Clb2:Cdc6 *Δcdc55* ****p = 0.0001 were calculated using unpaired Student's *t*-test).

The online version of this article includes the following figure supplement(s) for figure 2:

**Figure supplement 1.** Cdc6 suppression in *Δclb2* is through Cdc6 N-terminal phospho-degron.

## Clb2 stabilizes Cdc6 through its N-terminus

Clb2 binds to the Cdc6 N-terminus through a conserved LxF motif (*Örd et al., 2019*). We tested if low Cdc6 protein levels in *Δclb2* cells are mediated through the Cdc6 N-terminus. The *CDC6-T368A* mutant contains an alanine substitution in the C-terminal phospho-degron, stabilizing Cdc6 in mitosis (*Figure 2A*, lanes 4 + 5, *Figure 2B*; *Perkins et al., 2001*). This mutant retains the N-terminal Cdc6 phospho-degron at T39-S43. Cdc6-T368A was unstable in *Δclb2* log-phase cells (*Figure 2A*, lanes 2 + 3) and was more prominent in mitotic-arrested cells, supporting the idea that Clb2 shields the Cdc6 N-terminal phospho-degron (*Figure 2A*, lanes 5 + 6, *Figure 2B*; *Örd et al., 2019*). The *CDC6-T39A-T368A* double mutant, containing mutations at both the N- and C-terminal phospho-degron, was synthetically lethal in *Δclb2* cells (*Figure 2C*). When Cdc6 is overexpressed and controlled under the *GAL* promoter, we observed stabilized Cdc6-T39A-T368A in *Δclb2* cells during mitosis, suggesting that stabilized Cdc6 is lethal when Clb2's inhibitory role in origin licensing is abolished (*Figure 2—figure supplement 1*). In contrast, GAL-Cdc6 and GAL-Cdc6-T368A were degraded in *Δclb2* cells, which suggest the importance of the N-terminal phospho-degron in Cdc6 degradation when Clb2 is abolished (*Figure 2—figure supplement 1*).

## Cdc6–Clb2 interaction is enhanced in *Δcdc55* cells

Cdc6 stabilization in *Δcdc55* cells indicates that PP2A^Cdc55 inhibits Cdc6 protein accumulation. Therefore, we excluded the possibility that PP2A^Cdc55 targets neither Cdc6 phospho-degron. We hypothesized that PP2A^Cdc55 releases Clb2 from Cdc6, exposing the Cdc6 phospho-degrons, which subsequently leads to Cdc6 degradation. To test this possibility, we compared the Clb2:Cdc6 protein-binding ratio between wild-type and *Δcdc55* cells during mitosis by co-immunoprecipitation (co-IP). Cdc6 was stabilized in *Δcdc55* cells, confirming the results in *Figure 1B* (*Figure 2D*, lanes 3 + 6, *Figure 2E*). There was more Clb2 bound to Cdc6 in *Δcdc55* cells compared to wild-type cells (*Figure 2D*, lanes 3 + 6). The relative ratio between Clb2 to Cdc6 was reproducibly higher in *Δcdc55* cells, suggesting that PP2A^Cdc55 disrupts Cdc6–Clb2 binding (*Figure 2E*). From this result, we conclude that Clb2 binds tighter to a fraction of Cdc6 in *Δcdc55* cells, which may mask the Cdc6 N-terminal phospho-degron, and thereby contribute to Cdc6 stability in *Δcdc55* cells.

## PP2A^Cdc55 targets the Cdc6 phosphorylation site T7

Our finding supports a scenario in which PP2A^Cdc55 dephosphorylates Cdc6 at the N-terminus and disrupts Cdc6–Clb2 binding. We hypothesized that PP2A^Cdc55 dephosphorylates Cdc6 at residue T7, which removes the Cks1 docking site, thus inhibiting Clb2 binding. We generated two *CDC6-9MYC* constructs that contain alanine mutations at the Cdk1 phosphorylation sites to test if Cdc6-T7 site is dephosphorylated by PP2A^Cdc55. The *CDC6-6A* mutant strain contains six alanine mutations at T7A, T23A, T39A, S43A, T368A, and S372A, serving as a negative control. The *CDC6-T7* mutant strain only retains the T7 phosphorylation site while all other Cdk1 phosphorylation sites are mutated to alanine (T23A, T39A, S43A, T368A, and S372A) (*Figure 3A*). Thus, the *CDC6* constructs eliminate the function of both phospho-degrons. We raised a custom Cdc6-T7 phospho-specific antibody to detect T7 phosphorylation (Cdc6-T7p). Cdc6-T7 phosphorylation was not detected in the *CDC6-6A* mutant, confirming the specificity of the custom-made Cdc6-T7 phospho-specific antibody (*Figure 3B*). The Cdc6-6A protein level did not differ between *CDC55* and *Δcdc55* cells (*Figure 3B*). Cdc6-T7 phosphorylation was detected in the Cdc6-T7 construct and was slightly enhanced in *Δcdc55* cells, supporting the idea that PP2A^Cdc55 dephosphorylates the Cdc6-T7 site (*Figure 3B, C*). Cdc6-6A did not interact with Clb2, supporting a role for Cdc6 phosphorylation in protein binding (*Figure 3B*). Cdc6-T7 associated with Clb2, supporting our previous results that Cks1 docking to T7 is crucial for the Cdc6–Clb2 interaction (*Figure 3B*). The Cdc6-T7 and Clb2 interaction was enhanced in *Δcdc55* cells (*Figure 3B, C*).

The *CDC6-T7* mutant contains mutations on phospho-degrons. Why is the Cdc6-T7 protein level slightly stabilized in the absence of Cdc55? (*Figure 3B*). There is a possibility that the T7 site serves as a phospho-degron, because Clb2 may bind and mask phosphorylated Cdc6-T7 in *Δcdc55*. Interestingly, the Cdc6-6A protein level was noticeably less stable than Cdc6-T7 (*Figure 3B*). In order to understand the mechanism of these observations, we examined endogenous Cdc6, and the mutants Cdc6-6A and Cdc6-T7 protein levels in *cdc4-1* cells (*Figure 3—figure supplement 1*). Endogenous Cdc6 was stabilized after 1–2 hr at 37°C as reported previously (*Drury et al., 1997*; *Figure 3—figure*

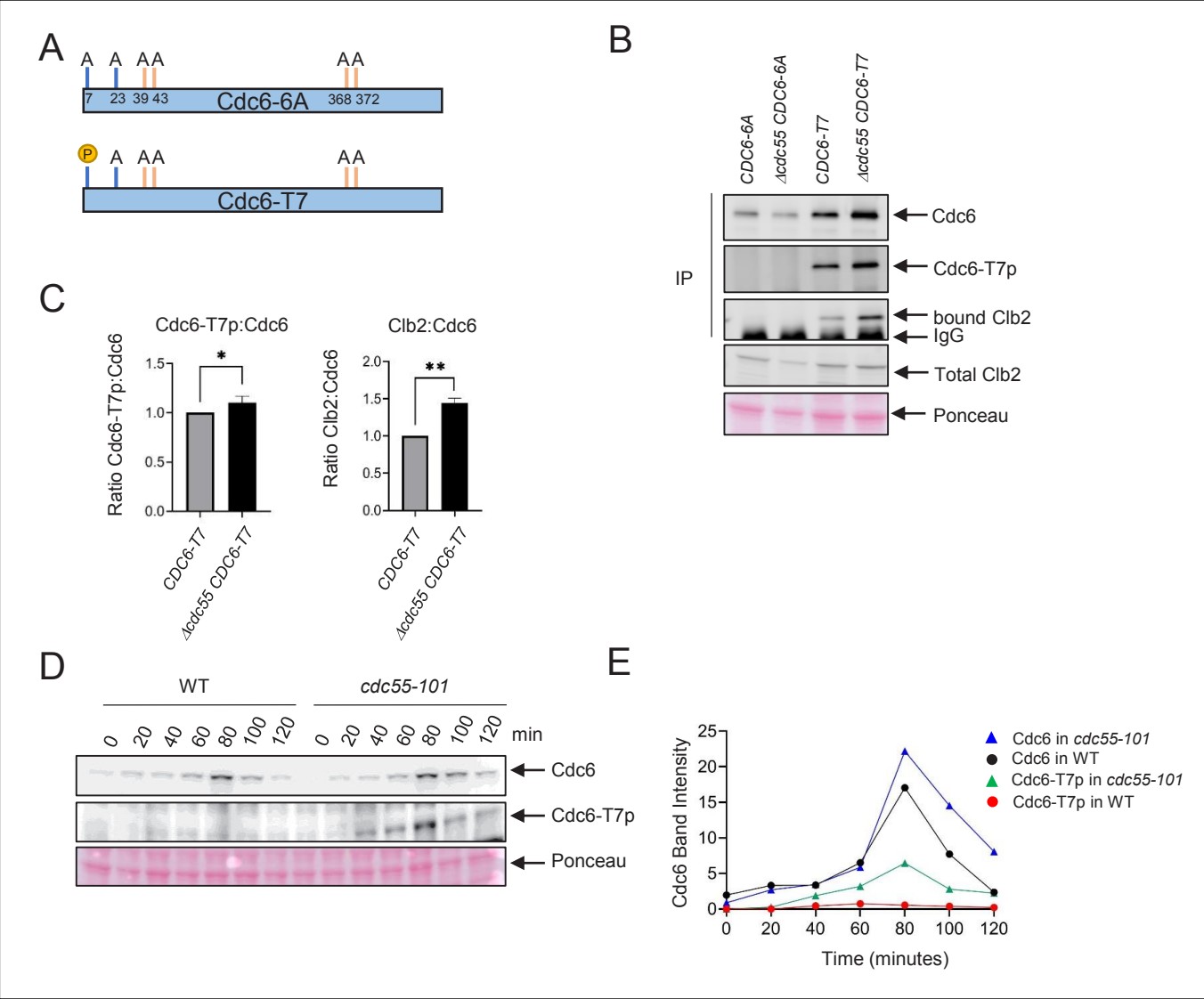

**Figure 3.** Cdc6-T7 is hyperphosphorylated in Δ*cdc55* cells. (**A**) A schematic of Cdc6 phosphorylation mutants is shown. Alanine mutations are indicated as A with amino acid positions. P indicates CDK sites accessible for phosphorylation. (**B**) The indicated Cdc6 mutants were immunoprecipitated using anti-MYC agarose beads. A total Cdc6 and Cdc6-T7 phosphorylation were analyzed by western blot with anti-MYC and Cdc6-T7p antibodies, respectively. Clb2 was detected by anti-Clb2. (**C**) Bands from B were quantified and the ratio between Cdc6-T7p:Cdc6 and Clb2:Cdc6 are shown. Error bars are standard error of the mean (SEM; *n* = 3; Cdc6-T7p:Cdc6 *p = 0.0468 and Clb2:Cdc6 **p = 0.0034 were calculated using unpaired Student's *t*-test). (**D**) *CDC6-9MYC* (*WT*) or *CDC6-9MYC cdc55-101* (*cdc55-101*) cells were arrested in G1 by α-factor and released. Cdc6 was visualized by anti-MYC antibody. Cdc6-T7 phosphorylation was detected by T7p phospho-specific antibody. (**E**) Band intensities from D were quantified.

The online version of this article includes the following figure supplement(s) for figure 3:

**Figure supplement 1.** Cdc6-6A is not targeted by SCF^Cdc4.

**Figure supplement 2.** WT and *cdc55-101* cells progress through the cell cycle.

*supplement 1*). In contrast, Cdc6-6A was still degraded in *cdc4-1* cells, indicating that Cdc6 has another degradation mechanism rather than SCF-mediated degradation. The Cdc6-T7 protein level stayed the same in *cdc4-1* cells, which indicates the SCF^Cdc4 does not target T7 for degradation (*Figure 3—figure supplement 1*).

Next, we examined Cdc6-T7 phosphorylation during the cell cycle by α-factor block and release. Cdc6 was expressed during mitosis at 80 minutes in wild type and was more stabilized in *cdc55-101* cells (*Figures 3D and 1D*). Cdc6-T7 phosphorylation was suppressed in wild-type cells (*Figure 3D*) throughout the cell cycle. In contrast, there was a spike of T7 phosphorylation in *cdc55-101* mutant

cells at 80 minutes during mitosis (*Figure 3D*). The Cdc6-T7 phosphorylation peak coincides with Cdc6 stabilization (*Figure 3D, E*). Flow cytometry analysis showed that both wild-type and *cdc55-101* mutant cells are in mitosis at 80 min (*Figure 3—figure supplement 2*). From these results, we conclude that Cdc6-T7 phosphorylation is reversed by PP2A$^{Cdc55}$, which releases Clb2.

## Cdc14 stabilizes Cdc6 in late mitosis

It has been reported that Cdc14 targets Cdc6 and other pre-RC proteins (*Zhai et al., 2010*). Consequently, we examined Cdc6 protein levels in *cdc14-3* mutant cells. Cdc14 was inactivated at the nonpermissive temperature upon α-factor release. We show that Cdc6 accumulation was consistently lower in *cdc14-3* compared to *cdc15-1*, an alternative mitotic mutant (*Figure 4—figure supplement 1A*). This result was especially prominent during mitosis at 120–180 minutes. The cell cycle profiles between *cdc14-3* and *cdc15-1* were the same, confirming that the observed difference in Cdc6 protein levels was not caused by cell cycle progression defects (*Figure 4—figure supplement 1B*). From these data, we conclude that Cdc14 stabilizes Cdc6 during late mitosis.

## PP2A$^{Cdc55}$ and Cdc14 dephosphorylate Cdc6 at distinct sites

PP2A$^{Cdc55}$ and Cdc14 exhibited opposite effects on Cdc6 stability (*Figure 1* and *Figure 4—figure supplement 1A*). We hypothesized that Cdc14 may target a Cdc6 phospho-degron, leading to Cdc6 stabilization. PP2A$^{Cdc55}$ and Cdc14 specificity toward Cdc6 was examined by in vitro phosphatase assays. Recombinant Cdc6 was preincubated with Clb5–Cdk1 in the presence of [γ-$^{32}$P]-ATP. The reaction mixture was further incubated with purified Cdc14 or PP2A$^{Cdc55}$ for dephosphorylation. Purified Sic1 was also added to the mixture in order to inhibit further phosphorylation of Cdc6 by Clb5–Cdk1 after addition of the phosphatases. The intensity of [$^{32}$P]-label on Cdc6 was measured by autoradiography. Full-length wild-type Cdc6 was moderately dephosphorylated by both PP2A$^{Cdc55}$ and Cdc14 (*Figure 4A*), indicating that Cdc6 is directly targeted by these phosphatases. The phosphate on Cdc6-T7 was efficiently removed by PP2A$^{Cdc55}$ when T7 was the only phosphorylation site available, but not by Cdc14 (*Figure 4B*). The same trend was observed with Cdc6-T23, but to a lesser extent (*Figure 4C*). When the Cdc6 C-terminal phospho-degron was the only phosphorylation site available, it was dephosphorylated by Cdc14, but not by PP2A$^{Cdc55}$ (*Figure 4D*). N-Swe1 and Csa1 were used as positive controls to show PP2A$^{Cdc55}$ *Harvey et al., 2011* and Cdc14 specificity (*Örd et al., 2020*), respectively (*Figure 4E, F*). Neither Cdc6-T39 nor -S43 was dephosphorylated by PP2A$^{Cdc55}$ or Cdc14, indicating that the Cdc6 N-terminal phospho-degron is not controlled by these phosphatases (*Figure 4—figure supplement 2*). These results suggest that PP2A$^{Cdc55}$ directly dephosphorylates Cdc6-T7 and -T23, while Cdc14 dephosphorylates the Cdc6 C-terminal phospho-degron at T368-S372. Thus, PP2A$^{Cdc55}$ and Cdc14 showed sequence specificity toward Cdc6.

## Sic1 releases Clb2 from Cdc6

Deletion of *CDC55* slows down S-phase progression. However, cells were still capable of completing S phase (*Figure 1F*). The key regulator of Clb2–Cdk1 inhibition, Sic1, is a promoter of origin licensing in G1 in conjunction with APC$^{Cdh1}$ (*Lengronne and Schwob, 2002*; *Wäsch and Cross, 2002*). It is possible that Sic1 also drives origin licensing through Cdc6. Both Cdh1 and Sic1 act in overlapping mechanisms that promote mitotic exit, which is potentially coupled to origin licensing (*Visintin et al., 1998*). We hypothesized that this is due to high level of Clb2 binding to Cdc6. A *CDC6-T39A-T368A Δcdh1* strain was used to obtain stable Cdc6 expression. Transient Sic1 expression was induced in a *GAL-SIC1* strain using galactose. Overexpression of Sic1 released Clb2 from Cdc6 in mitotic cells arrested by nocodazole (*Figure 5A, B*).

Next, we tested if Sic1 promotes Mcm2–7 DNA loading using a biochemical approach with purified proteins (*Ayuda-Durán et al., 2014*; *Remus et al., 2009*; *Seki and Diffley, 2000*). The stringent regulation of intracellular Cdc6 protein levels by proteolytic degradation in both G1 and S phase severely hampers the purification of endogenous Cdc6 from these cell cycle stages even after galactose-induced overexpression. However, we observed that extended overexpression of TAP$^{TCP}$-tagged Cdc6 upon release from α-factor arrest induces a mitotic arrest phenotype in yeast cells, characterized by a 2C DNA content and the presence of a large elongated bud (data not shown). Intriguingly, Cdc6 isolated from whole-cell extracts of such mitotically arrested cells exists in a stable stoichiometric complex with three additional proteins that we identified as Clb2, Cdk1, and Cks1 by mass

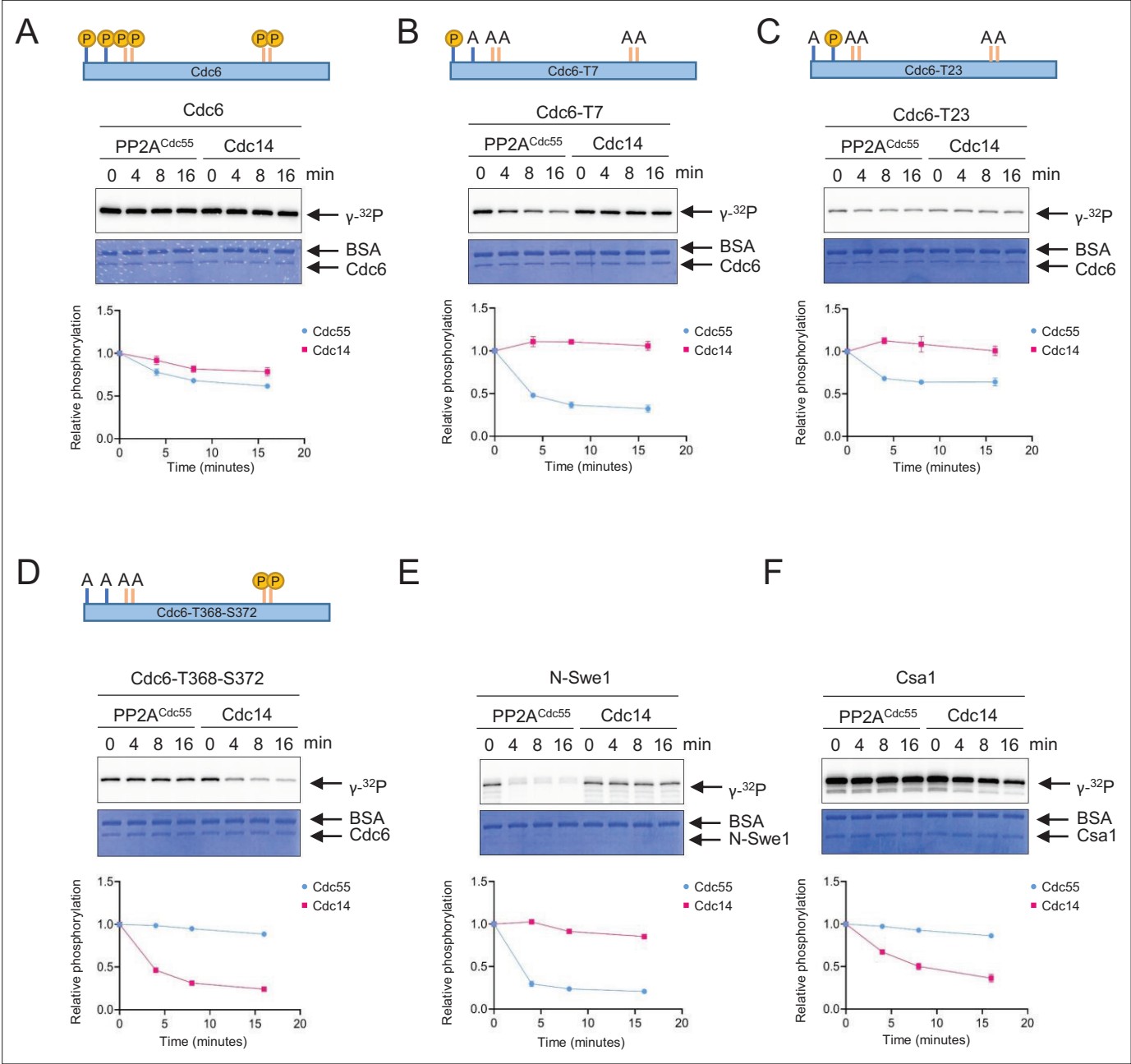

**Figure 4.** PP2A and Cdc14 dephosphorylate Cdc6-T7, -T23, and T368-S372, respectively. Full-length Cdc6 (**A**), Cdc6-T23A-T39A-S43A-T368A-S372A (Cdc6-T7) (**B**), Cdc6-T7A-T39A-S43A-T368A-S372A (Cdc6-T23) (**C**), and Cdc6-T7A-T23A-T39A-S43A (Cdc6-T368-S372) (**D**) were phosphorylated for 60 min at room temperature using purified Clb5–Cdk1 in the presence of [γ-$^{32}$P]-ATP. At time 0, Sic1 and phosphatase Cdc14 or PP2A$^{Cdc55}$ were added to the reactions to inhibit Clb5–Cdk1. A time course was taken, and samples were measured for dephosphorylation. N-Swe1 was used as a positive control for PP2A$^{Cdc55}$ (**E**), and Csa1 for Cdc14 (**F**). The samples were loaded on sodium dodecyl sulfate–polyacrylamide gel electrophoresis (SDS–PAGE) and following electrophoresis, the gels were stained using Coomassie Brilliant Blue. Signals were quantified using ImageQuant TL. Three independent biological replicates of the phosphatase assays were performed. Error bars are standard error of the mean (SEM; $n$ = 3).

The online version of this article includes the following figure supplement(s) for figure 4:

**Figure supplement 1.** Cdc6 is unstable in *cdc14-3* mutant.

**Figure supplement 2.** PP2A and Cdc14 do not target Cdc6-T39 and S43.

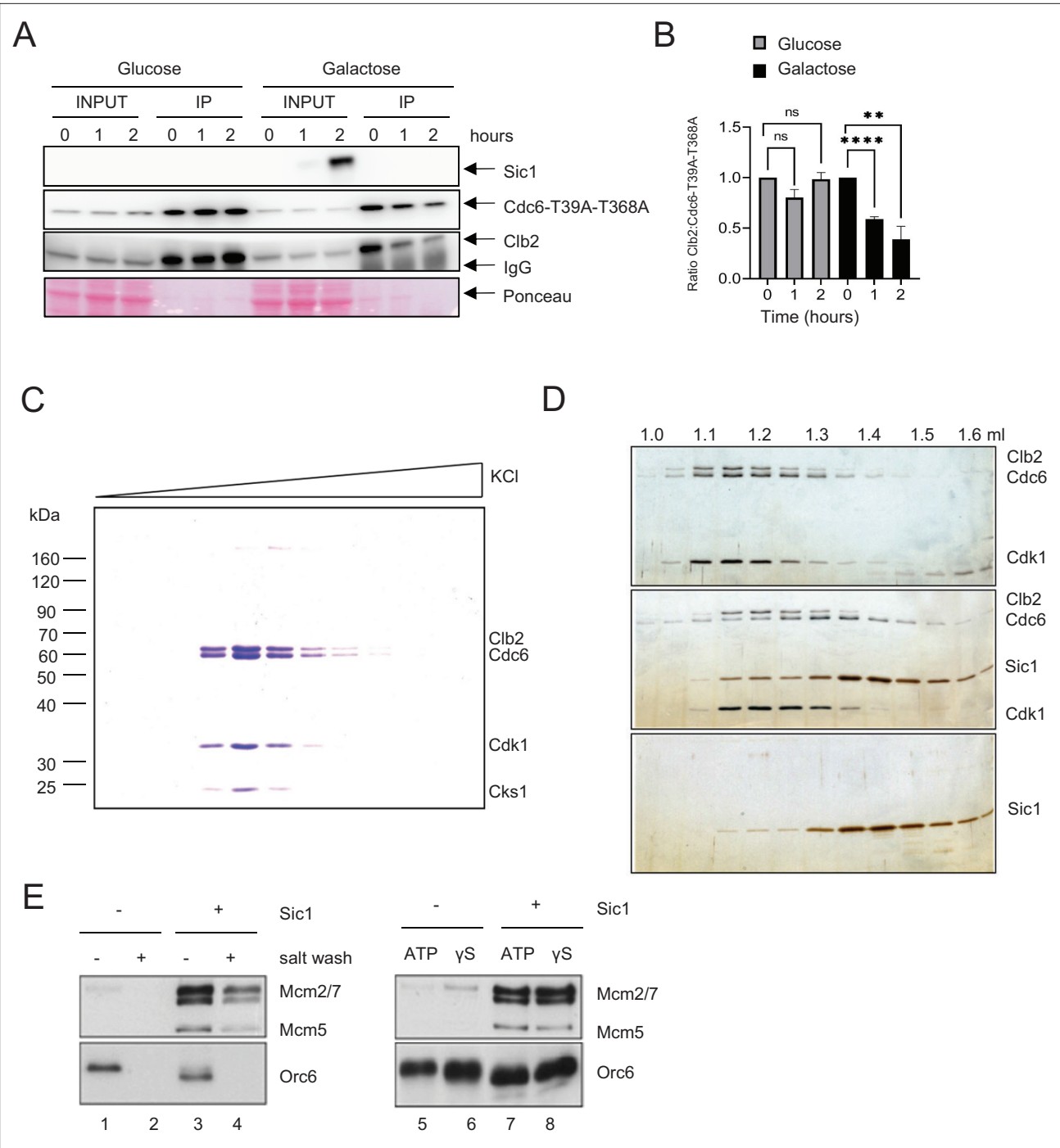

**Figure 5.** Sic1 promotes origin licensing by releasing Clb2–CDK from Cdc6. (**A**) *GAL-SIC1-HA CDC6-T39A-T368A-PrA Δcdh1* cells were arrested in mitosis by nocodazole for 2 hr before the addition of glucose or galactose. Samples were collected at the indicated time points and protein extracted (INPUT). Cdc6-T39A-T368A-PrA pulled down by anti-IgG Dynabeads was analyzed by western blot to visualize Sic1, Cdc6-T39A-T368A, and Clb2. (**B**) Clb2:Cdc6-T39A-T368A IP ratio relative to time 0 from A is shown. Error bars are standard error of the mean (SEM; *n* = 3; Glucose p = 0.8414, p = 0.0729, ns, not significant; Galactose ****p < 0.0001, **p = 0.0090 was calculated using unpaired Student's *t*-test). (**C**) Cdc6 was expressed and purified from M-phase cell extracts. The elution profile of the Cdc6·Clb2·Cdk1·Cks1 complex after ion-exchange chromatography is shown. (**D**) Purified Cdc6·Clb2·CDK complex and Sic1 were analyzed by gel filtration. Fractions were analyzed by sodium dodecyl sulfate–polyacrylamide gel electrophoresis (SDS–PAGE) and silver staining. Elution positions are indicated on the top. Gel positions of indicated proteins are located on the right. (**E**) Mcm2-7 loading assays with purified ORC, Cdc6·Clb2·Cdk1·Cks1, Cdt1·Mcm2–7, and Sic1 on origin DNA immobilized on paramagnetic beads. Loading was performed in the presence or absence of Sic1 and the presence of either a salt wash (left) or ATP and ATP analog, ATPγS (γS) (right).

spectrometry (*Figure 5C*). The formation of this complex after Cdc6 overexpression is consistent with the previous observation of a Cdc6·Clb2·Cdk1 complex in G2/M cells (*Mimura et al., 2004*). Importantly, using gel-filtration analysis, we find that Cdc6 is released from the Clb2–Cdk1 complex in the presence of the Cdk1 inhibitor Sic1, which itself forms a stable complex with Clb2–Cdk1 (*Figure 5D*).

Clb2–Cdk1 binding to Cdc6 has been proposed to inhibit the origin licensing activity of Cdc6 as part of cellular re-replication control mechanisms (*Mimura et al., 2004*). We tested the ability of Cdc6·Clb2·Cdk1·Cks1 to promote Mcm2–7 loading onto DNA in conjunction with purified ORC and Cdt1·Mcm2–7, analogous to experiments described previously with purified Cdc6[52]. Using this approach, we find that Cdc6·Clb2·Cdk1·Cks1 does not support the formation of salt-stable MCM complexes on DNA, indicating that Cdc6 in complex with Clb2·Cdk1·Cks1 is inactive for Mcm2–7 loading (*Figure 5E*, lanes 1 + 2). Importantly, normal Mcm2–7 loading was observed in reactions containing both purified Sic1 and Cdc6·Clb2·Cdk1·Cks1 (*Figure 5E*, lanes 3 + 4), indicating that Clb2·Cdk1·Cks1 sequesters Cdc6 and that this form of Cdc6 inhibition can be reversed by release of Cdc6 from the Cdc6·Clb2·Cdk1·Cks1 complex by Sic1. However, we noted that Orc6 is efficiently phosphorylated in the presence of Cdc6·Clb2·Cdk1·Cks1, as indicated by the pronounced gel-mobility shift of Orc6 (*Figure 5E*, compare lanes 1 + 3). This Orc6 phosphorylation is inhibited in the presence of Sic1, indicating that Clb2–Cdk1 retains partial kinase activity in the Cdc6·Clb2·Cdk1·Cks1 complex. It is known that Cdk1 phosphorylation inhibits the Mcm2–7 loading function of ORC (*Frigola et al., 2013*). It was possible that the inhibition of Mcm2–7 loading in the presence of Cdc6·Clb2·Cdk1·Cks1 was due to ORC phosphorylation and not Cdc6 sequestration. However, we noted that not only Mcm2–7 loading, but even Mcm2–7 recruitment to ORC/Cdc6, which can be monitored in the presence of ATPγS, is inhibited in the presence of Cdc6·Clb2·Cdk1·Cks1 (*Figure 5E*, lanes 5 + 6), and this inhibition is again reversed in the presence of Sic1 (*Figure 5E*, lanes 7 + 8). Since ORC phosphorylation specifically inhibits Mcm2–7 loading but not Mcm2–7 recruitment (*Frigola et al., 2013*), these data demonstrate that the mechanism of licensing inhibition by Cdc6·Clb2·Cdk1·Cks1 complex formation is distinct from that by Cdk1 phosphorylation of ORC. In summary, we conclude that Clb2–Cdk1 inhibits Cdc6 by physical sequestration and this inhibition is reversible by the Sic1-mediated release of Cdc6 from the Cdc6·Clb2·Cdk1·Cks1 complex.

## Key functional residues/motifs of Cdc6 and Clb2 are located within regions of intrinsic disorder

The necessary structural elements in Cdc6 responsible for protein binding are not well understood. Protein-binding motifs often fall in IDRs, which are not well characterized structurally due to the limitations of crystal structure analysis (*Wright and Dyson, 2015*; *Fuxreiter et al., 2007*). IDRs in Cdc6 were identified by several programs including IUPred (*Mészáros et al., 2018*) and Anchor (*Mészáros et al., 2009*), which include: (1) Cdc6-N terminus residues 1–59 and (2) Cdc6 C-terminus residues 350–378 (*Figure 6A* and *Figure 6—figure supplement 1A*). Next, we constructed a full-length Cdc6 structure model by extracting the known structure from Protein Data Bank (PDB ID: 5V8F) (*Yuan et al., 2017*), which was lacking some of the key functional residues and regions, for example, 1–59, 68–80, 129–163, and 349–387. Indeed, these regions fall into the IDRs identified in *Figure 6A*. We generated a complete structural representation of Cdc6 through predictions of both IDRs and secondary structure combined with template-based modeling and ab initio approaches (see Methods). The missing regions modeled in the Cdc6 N-terminus span residues 1–59 (*Figure 6B*, left) as well as the centrally located residues 350–378 (*Figure 6B*, right), which correspond to the predicted IDRs (*Figure 6A*). The Cdc6 N-terminal IDR comprises four Cdk1 consensus sites (T7, T23, T39, and S43) and two cyclin-binding motifs RxL (30–32) and LxF (47–49), all of which are critical for Cdc6 phospho-regulation (*Figure 6B*, left). The Cdc6 C-terminal IDR (residues 350–378) contains phospho-degron 368–372 (*Figure 6B*, right). We also compared our Cdc6 structural model with the recently released AlphaFold and evaluated the model quality using Verify 3D software (*Figure 6—figure supplement 1B*). The 3D-1D profiles from Verify 3D measure the compatibility of the primary structure of a protein with its three-dimensional structural environment in the model. An average 3D-1D score of 0.2 (shown by the green line in the plots) denotes the threshold of acceptable model quality, that is, scores around this threshold or above are inferred as correctly modeled regions (*Figure 6—figure supplement 1C*). The phospho-degrons (red boxes) scored close to or at the 0.2 threshold in our model, while both regions score well below the threshold in the AlphaFold model (*Figure 6—figure supplement 1C*). This is

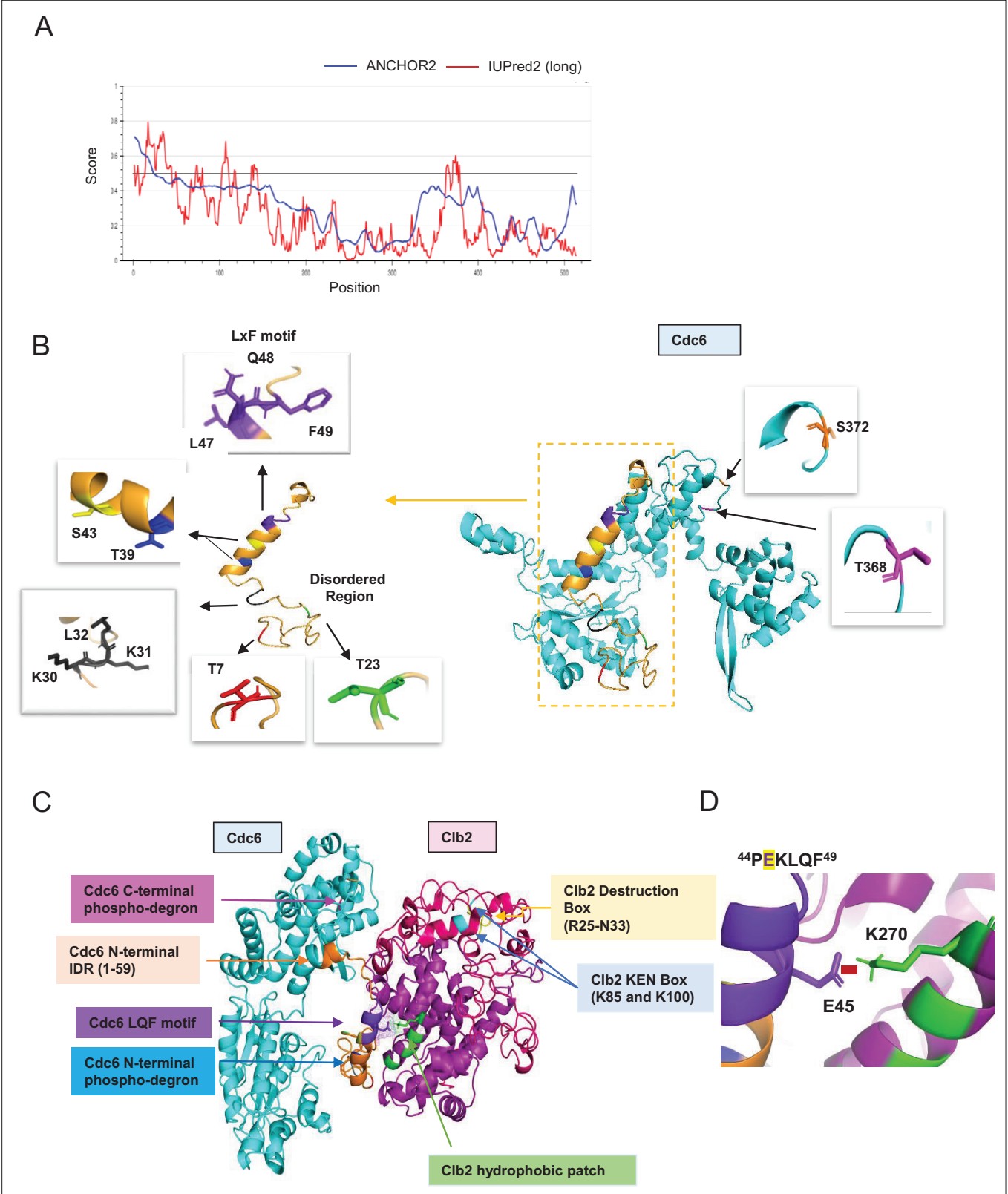

**Figure 6.** Predicted Cdc6–Clb2 structure by bioinformatics. (**A**) Disordered regions were mapped using ANCHOR2 and IUPred2 software. (**B**) A full-length Cdc6 protein structure composed of 513 amino acids was predicted based on the amino acid sequence and known structure information (cyan and orange on the right). N-terminal highly disordered region spans in 1–59 residues (orange box and highlighted on the left). T7 is in red, T23 is in green, RxL motif is in black, and LxF motif in purple. N-terminal phospho-degron is located at T39-S43 and C-terminal phospho-degron is at T368-S372.

*Figure 6 continued on next page*

*Figure 6 continued*

(**C**) Cdc6–Clb2-binding structure with Cdc6 IDR (orange), Cdc6 LQF motif (purple), Cdc6 N-terminal phospho-degron (blue), Cdc6 C-terminal phospho-degron (pink), Clb2 D-box (yellow), Clb2 KEN Box (light blue), and Clb2 hydrophobic patch (green). (**D**) Cdc6–Clb2 interaction through Cdc6 E45 (purple) and Clb2 K270 (green). Red bar shows salt bridge.

The online version of this article includes the following figure supplement(s) for figure 6:

**Figure supplement 1.** A prediction of intrinsically disordered regions (IDRs) in Cdc6.

also consistent with low confidence scores (less than 70) generated by AlphaFold's internal confidence rating (*Figure 6—figure supplement 1B*, yellow and orange). The scores in the structured regions of both models are comparable. Thus, our model outperforms the AlphaFold model in the region of the modeled IDRs (*Figure 6—figure supplement 1C*).

Finally, we predicted a potential scenario of Cdc6–Clb2 interaction, where Cdc6 E45 interacts with Clb2 K270 in hp via a salt bridge (*Figure 6C, D*) that highlights the importance of the IDR in this interaction. We speculate that a transiently structured helix adopted by the PEKLQF motif in our Cdc6 model may be important for its interaction with the hp in Clb2. The N-terminus of Clb2 located from residues 1–210 (data not shown) contained an IDR which includes the KEN box (residues 85 and 100) and the destruction box (D-box) (residues 25–33), which are both necessary for Clb2 degradation via APC (*Pfleger and Kirschner, 2000*; *Schwab et al., 1997*; *Glotzer et al., 1991*; *Figure 6C*). The ordered regions of Clb2 include the hp (N260, L264, Y267, and K270) that facilitates protein binding to its substrate such as Cdc6 (*Örd et al., 2019*; *Hunt, 1991*; *Figure 6C*). Importantly, our Cdc6–Clb2 protein interaction model also captured the Cdc6 N-terminal phospho-degron protected by Clb2, thereby inhibiting Cdc6 degradation.

## Discussion

### Cdc6 regulation by PP2A^Cdc55

In this study, we show that PP2A^Cdc55 dephosphorylates Cdc6 at T7 and T23 to disrupt the Cdc6–Clb2 complex in mitosis. As cells progress into late mitosis, Sic1 releases Clb2·Cdk1·Cks1 from Cdc6 to promote origin licensing. Following this, Cdc14 dephosphorylates Cdc6 at T368-S372 to stabilize Cdc6 before origin licensing. Our bioinformatical analyses demonstrate that highly adaptable IDRs in the Cdc6-N terminus are critical for Cdc6-binding dynamics. The data provide a temporal model of origin licensing control by Clb2·Cdk1·Cks1 release and sequential dephosphorylation of Cdc6 (*Figure 7A*).

Neither PP2A^Cdc55 nor Cdc14 dephosphorylated the Cdc6 N-terminal phospho-degron, suggesting that there are additional mechanisms involved (*Figure 4—figure supplement 2*). The Cdc6-T7 site is located in an IDR, where PP2A^Cdc55 and Cks1 might gain easier access for docking (*Figure 6A, B*), which may trigger a cascade of Cdc6 dephosphorylation in early mitosis (*Touati et al., 2019*). Our study also supports a model in which PP2A^Cdc55 prefers phospho-threonines over phospho-serines (*Godfrey et al., 2017*).

Previous work in *Xenopus* egg extracts indicated that immunodepletion of the trimeric PP2A complex inhibits replication initiation without regulating pre-RC assembly (*Lin et al., 1998*). A follow-up study suggested that PP2A targets proteins required for DNA replication initiation but not elongation proteins, for example, Cdc45 (*Chou et al., 2002*). Sld2 and Sld3 are potential substrates for PP2A (*Bloom and Cross, 2007*). Therefore, PP2A might regulate multiple replication proteins at different steps. Mammalian PP2A is regulated by a diverse group of regulatory subunits, termed, B, B', B", and B"' (*Janssens and Goris, 2001*). B55 is a mammalian homolog of Cdc55 which plays a role in mitosis (*Schmitz et al., 2010*). PP2A^B55 also selectively targets phospho-threonines over phospho-serines in vitro (*Cundell et al., 2016*; *Agostinis et al., 1990*). PR48 (B" family) has been identified as a Cdc6-binding partner through yeast two-hybrid screening (*Yan et al., 2000*). PR48 mediates protein interaction with the Cdk1 phosphorylation site-containing Cdc6 N-terminus (*Yan et al., 2000*). PR70, another B" family subunit, has also been implicated in Cdc6 regulation by enhancing phosphatase activity toward Cdc6 (*Yan et al., 2000*; *Wlodarchak et al., 2013*; *Davis et al., 2008*). Thus, our data define Cdc6 dephosphorylation by PP2A in a simple system, which might be a conserved mechanism in eukaryotes.

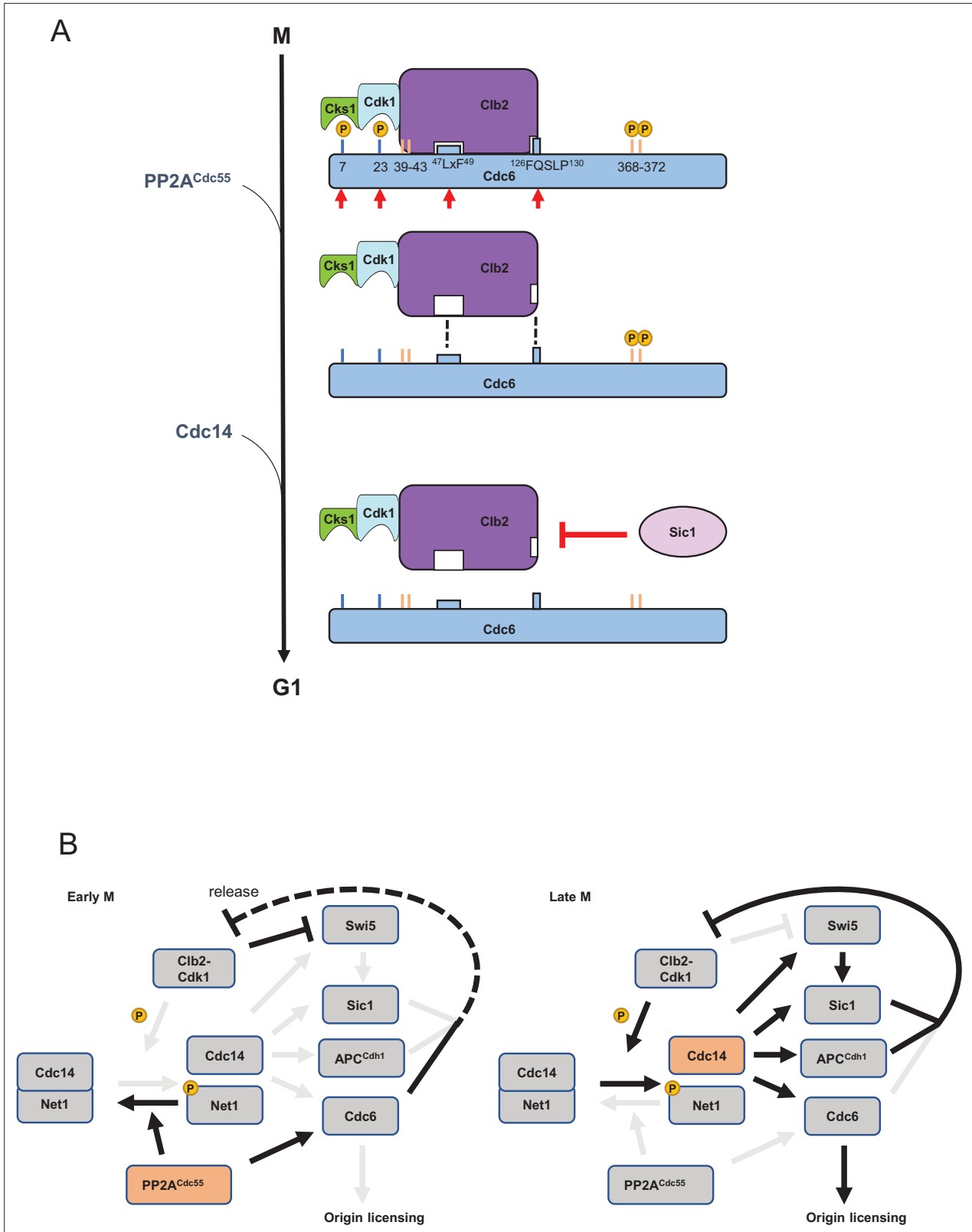

**Figure 7.** A model of Cdc6 regulation by Clb2, PP2A[Cdc55], Sic1, and Cdc14. (**A**) In mitosis, the Clb2·Cdk1·Cks1 complex binds to phosphorylated Cdc6 N-terminus to mask the phospho-degron and prohibit Cdc6 loading on DNA, which is mediated by the LxF motif and FQSLP enhancer region. PP2A[Cdc55] dephosphorylates Cdc6-T7 and -T23 sites to disrupt Cdc6·Clb2·Cdk1·Cks1 complex. Sic1 releases Clb2·Cdk1·Cks1 from Cdc6 to promote origin licensing. Cdc14 dephosphorylates Cdc6 C-terminal phospho-degron for stabilization. Cks1 docking sites T7 and T23 are colored blue whereas

*Figure 7 continued on next page*

Figure 7 continued

phospho-degrons are colored orange. Red arrows show sites and motifs where Clb2–Cdk1 inhibits Cdc6 loading. Dashed lines indicate reduced binding. (**B**) In early mitosis, PP2A$^{Cdc55}$ inhibits Cdc14 release and promotes a rise in Clb2–Cdk1 activity by dephosphorylating Cdc6. During late mitosis, Clb2–Cdk1 initiates Cdc14 release from inhibitory binding by Net1. Cdc14 targets Swi5, Sic1, Cdh1, and Cdc6, contributing to full suppression of Clb2–Cdk1 activity and origin licensing.

## Cdc14 and DNA replication

The Cdc6 C-terminal phospho-degron is targeted by SCF$^{Cdc4}$ for protein degradation in mitosis (**Drury et al., 2000**). Here, we show that the Cdc6 C-terminal phospho-degron is dephosphorylated by Cdc14 to stabilize Cdc6 (**Figure 4**). Cdc14 is inhibited by PP2A$^{Cdc55}$ through Net1 dephosphorylation (**Queralt et al., 2006**), which indicates that Cdc14-dependent Cdc6 dephosphorylation is not triggered until Clb2–Cdk1 activity overrides PP2A$^{Cdc55}$ activity. Elevated Clb2–Cdk1 activity phosphorylates Net1 to release Cdc14 from nucleolus (**Figure 7B**). PP2A$^{Cdc55}$-dependent Cdc6-T7 dephosphorylation releases Clb2, which may enhance Net1 phosphorylation because Clb2 is free from its inhibitor Cdc6 (**Figure 7B**, left). Once Cdc14 is released from Net1, Cdc14 dephosphorylates Cdc6, Cdh1, and Sic1 to license the origin (**Figure 7B**, right). This temporal regulation of Cdc14 ensure that Cdk1 substrates are sequentially dephosphorylated (**Kataria et al., 2018**). Such a temporal regulation is also achieved by cell cycle-dependent Cdc55 localization; nuclear Cdc55 inhibits the metaphase–anaphase transition through APC inhibition (**Rossio et al., 2013**; **Queralt and Uhlmann, 2008**).

There is evidence that Cdc14 is a key regulator for promoting DNA replication. When Cdc14 is not properly sequestered in *CDC14-TAB6* mutant, cells show DNA replication defects when *CLB5* is deleted (**Bloom and Cross, 2007**). Cdc14 dephosphorylates replication initiation proteins Orc2, Orc6, Cdc6, and Mcm3 (**Zhai et al., 2010**). Ectopic Cdc14 expression induces DNA re-replication, supporting the idea that Cdc14-dependent dephosphorylation of pre-RC components is a requirement for origin licensing (**Zhai et al., 2010**). Thus, Cdc14 is a key component to assemble pre-RCs in late mitosis.

## Clb2 function in Cdc6

Our results show that Clb2·Cdk1·Cks1 binding to Cdc6 may hinder SCF$^{Cdc4}$ recognition of the N-terminal phospho-degron and consequently leads to Cdc6 stabilization (**Figures 2 and 6C**). This indicates that the Cdc6 C-terminal phospho-degron is the key element for Cdc6 degradation until Clb2 is released. We have previously shown that the Cdc6 C-terminal phospho-degron is targeted by Mck1, a yeast homolog of GSK3, for protein degradation (**Al-Zain et al., 2015**; **Ikui et al., 2012**). Despite high Clb2–Cdk1 activity and an exposed phospho-degron, Cdc6 can be partially stabilized in mitosis until Mck1 phosphorylates Cdc6-T368. It is of interest to study how Mck1 is activated during the cell cycle. Mck1 is involved in DNA damage signaling (**Al-Zain et al., 2015**; **Li et al., 2019**), therefore the Cdc6 C-terminus, which is available throughout the cell cycle, plays a role in controlling the timing of licensing under stress conditions. We also showed evidence that Cdc6 is degraded in SCF$^{Cdc4}$-independent manner; for example, Cdc6-6A mutant is unstable despite the mutations at phospho-degrons (**Figure 3—figure supplement 1**).

APC$^{Cdh1}$ targets Clb2 for degradation via the KEN- and D-boxes (**Pfleger and Kirschner, 2000**; **Schwab et al., 1997**; **Glotzer et al., 1991**). Cdh1 is inhibited until it is dephosphorylated by Cdc14. The KEN- and D-boxes in Clb2 are located in IDRs, which may also enhance interaction with the APC to facilitate Clb2 degradation upon mitotic exit (**Figure 6C**; **Radivojac et al., 2010**).

## Sic1 regulation

We showed that Sic1 releases Clb2 from Cdc6 which supports origin licensing (**Figure 5**). Sic1 contains multiple Cdk1 phosphorylation sites, which includes a phospho-threonine Cks1 docking site in the N-terminus (**Kõivomägi et al., 2011**; **Venta et al., 2020**). Cks1 docking triggers an N- to C-directed multisite phosphorylation cascade both in Sic1 and Cdc6, which results in the creation of phospho-degrons targeted by SCF$^{Cdc4}$ [14,18]. Thus, Cdc6 and Sic1 share a functional similarity in their phosphorylation patterns. Since PP2A$^{Cdc55}$ dephosphorylates Cks1 docking site T7 in Cdc6, it is worth examining if PP2A$^{Cdc55}$ dephosphorylates Sic1 at the N-terminal Cks1 docking site. Sic1, Cdh1, and Cdc6 are Clb2–Cdk1 inhibitors as well as Cdc14 substrates and SCF$^{Cdc4}$ targets (**Örd et al., 2019**; **Calzada et al., 2001**; **Visintin et al., 1998**; **Verma et al., 1997**). Cdc14-dependent Swi5 dephosphorylation

also activates Sic1 nuclear import and transcription at the end of mitosis (*Visintin et al., 1998*). Thus, Cdc14 release from the nucleolus targets at least four components to achieve a complete inhibition of Clb2–Cdk1 activity during mitotic exit, leading to origin licensing (*Figure 7B*).

### Short linear motifs and IDRs

Recent studies have expanded the role of cyclin-specific short linear motifs (SLiMs), which are short functional sequence motifs often found in IDRs. SLiMs such as RxL and LxF motifs dictate Cdk1 substrate specificity and control cell cycle progression (*Örd et al., 2019*; *Örd et al., 2020*; *Faustova et al., 2021*; *Bhaduri and Pryciak, 2011*). This is also conserved in mammalian cells (*Takeda et al., 2001*; *Allan et al., 2020*; *Topacio et al., 2019*). Phosphatases also utilize SLiMs for substrate docking. In yeast, Cdc14 scaffolds to a PxL motif in substrates to mediate dephosphorylation events (*Kataria et al., 2018*). Currently, there is no identified docking motif for PP2A$^{Cdc55}$ in yeast. Mammalian homolog PP2A$^{B55}$ recognizes a polybasic docking sequence that flanks CDK phosphorylation sites (*Cundell et al., 2016*). PP2A$^{B56}$-specific SLiMs are defined as LSPIxE in humans (*Wang et al., 2016*). Future research should focus on docking motifs for phosphatases and their regulatory subunits to understand phosphatase specificity in yeast.

Our data illustrate multiple IDRs in Cdc6 (*Figure 6A*). Notably, the N-terminal Cdc6 IDR includes SLiMs such as RxL and LxF motifs, which supports a model that depicts the Cdc6 N-terminus as a hub for multiple protein–protein interactions (*Figure 6B*). Cdc6 IDRs are readily available for protein binding in G1/S, during which Cdc6 disengages from ORC and is subjected to SCF-dependent degradation (*Drury et al., 1997*; *Piatti et al., 1995*). Cdc6 is then shifted toward an origin licensing preparation period where Cdc6 is stabilized but not licensed via Clb2 binding. This temporal mode is released when Cdc14 outcompetes with Clb2–Cdk1 activity at the end of mitosis.

In humans, cyclin A–CDK2 was shown to utilize SLiMs to promote interaction with Cdc6 and ORC (*Hossain et al., 2021*). Our studies on Cdc6 regulation shed light on the dynamics of phosphorylation and dephosphorylation that drive DNA replication initiation upon mitotic exit. In this study, we showed first evidence to link PP2A with DNA replication. This study also provided a stepwise origin licensing process mediated by PP2A, Cdc14, and Sic1.

## Methods

### Yeast cultures

Yeast extract peptone medium with glucose (YPD) was used to grow cells for western blot and flow cytometry analysis. Low fluorescence medium with glucose supplemented with adenine was used to visualize Cdc6-GFP under fluorescence microscope (*Sheff and Thorn, 2004*). G1 arrest was achieved by α-factor at 50 nM for 2 hr at 30°C. Cells were washed three times and resuspended in YPD to release the G1 arrest. Mitotic arrest was achieved by nocodazole at 15 µg/ml for 2 hr at 30°C.

### Yeast strains

Standard methods were used for mating, tetrad analysis, and transformation. All yeast strains are haploid congenic to W303 background. Strain list in this study is in *Supplementary file 1*.

### Western blotting and Co-IP

Cells were lysed in TBT buffer containing protease inhibitors and Phos STOP (Roche) with acid-washed glass bead agitation using Fast-Prep as previously described (*Bloom et al., 2011*). Proteins were separated by sodium dodecyl sulfate–polyacrylamide gel electrophoresis (SDS–PAGE) with Novex 4–20% Tris–glycine polyacrylamide gel (Life Technologies). Cdc6-9MYC was detected using anti-cMYC antibody 9E10 (Sigma-Aldrich Cat# M4439, RRID:AB_439694) at 1:5000 dilution and anti-Pgk1 (Thermo Fisher Scientific Cat# 459250, RRID:AB_2532235) at 1:5000 as a loading control. Cdc6-protein A was detected by HRP-conjugated rabbit IgG (Sigma-Aldrich Cat# P1291, RRID:AB_1079562). Anti-Clb2 antibody is a gift from Frederick Cross. Images were developed using a Fuji LAS 4000 Imager (GE Healthcare Life Sciences, Pittsburgh, PA) or Odyssey CLx imager (LI-COR Lincoln, NE). Co-immunoprecipitation was performed with rabbit anti-MYC-conjugated agarose beads (Sigma-Aldrich, St Louis, MO) for 1 hr at 4°C. Cdc6-T7 phospho antibody is custom antibody raised in rabbit (Covance,

Princeton, NJ). Sic1-HA and Cdc6-HA were detected by HRP-conjugated rat HA (Roche Cat# 12013819001, RRID:AB_390917).

## FACS

Cells were fixed and stained with propidium iodide staining as previously described (*Epstein and Cross, 1992*). Flow cytometry analysis was performed using a BD Accuri C6 flow cytometer (BD Biosciences). A total of 20,000 cells were analyzed per sample. FL2 channel was used to detect the fluorescent signal.

## Protein purification for in vitro phosphatase assay

Cdc6, Sic1, Swe1(1–450), and Cdc14 were expressed in *E. coli* BL21RP cells from pET28a-based vectors as fusions with 6xHis tag. Csa1 was expressed as GST fusion in BL21RP cells from pGEX-4T1-based vector. The expression of Cdc6 and Swe1(1–450) was induced at 23°C using 0.3 mM IPTG, Sic1 at 37°C using 1 mM IPTG, Cdc14 at 23°C using 0.125 mM IPTG, and Csa1 at 16°C using 0.3 mM IPTG. 6xHis-Cdc6, 6xHis-Sic1, and 6xHis-Swe1(1–450) were purified by standard cobalt affinity chromatography with 200 mM imidazole used for elution. 6xHis-Cdc14 was purified using nickel affinity chromatography with 250 mM imidazole used for elution. GST-Csa1 was purified using Glutathione Sepharose (GE Healthcare). Clb5–Cdk1 complex was purified from *S. cerevisiae* culture where TAP-tagged Clb5 was overexpressed from *GAL1* promoter as described previously (*Puig et al., 2001*; *Ubersax et al., 2003*). The yeast lysate was prepared using Mixer Mill MM 400 (Retch). PP2A$^{Cdc55}$ complex was purified from yeast cells containing 3HA-tagged Cdc55 (*Anastasia et al., 2012*). The yeast culture was grown in YPD to $OD_{600}$ = 1.4, when the cells were collected and snap frozen. The cells were lysed using Mixer Mill MM 400. The lysate was cleared by centrifugation and the supernatant was incubated with anti-HA agarose beads for 3 hr. The beads were washed thoroughly and PP2A$^{Cdc55}$ was eluted with buffer containing HA dipeptide.

## In vitro dephosphorylation assay

Substrate proteins Cdc6, Swe1(1–450), and Csa1 at 500 nM concentration were phosphorylated for 60 min at room temperature using 2 nM Clb5–Cdk1 in a buffer containing 50 mM HEPES–KOH, pH 7.4, 150 mM NaCl, 5 mM $MgCl_2$, 20 mM imidazole, 2% glycerol, 0.2 mg/ml BSA, and 500 µM ATP [(with added [γ-$^{32}$P]-ATP (Hartmann Analytic))]. Prior to addition of phosphatase mix, an aliquot of the reaction was pipetted to SDS sample buffer. Then, a mixture containing Sic1 and phosphatase Cdc14 or PP2A$^{Cdc55}$ was mixed with the phosphorylation reactions to inhibit Clb5–Cdk1 and to measure the dephosphorylation. The final concentration of Sic1 was 1 µM. At 4, 8, and 16 min, an aliquot of the reaction was mixed with SDS sample buffer to stop the reaction. The samples were loaded on SDS–PAGE and following electrophoresis, the gels were stained using Coomassie Brilliant Blue R-250 dye and dried.

Γ-$^{32}$P phosphorylation signals were detected using an Amersham Typhoon 5 Biomolecular Imager (GE Healthcare Life Sciences). Signals were quantified using ImageQuant TL (Image Quant TL, RRID:SCR_018374). The dephosphorylation assays were performed in three independent replicate experiments.

## Microscope

Images were acquired with Nikon Eclipse Ti2 microscope with ×60 Oil CFI Plan APO Lambda lens. Fluorescence illumination is provided by Sola light engine and images were captured by Nikon DS-Qi2. Images were acquired by Nikon NIS element software.

## Time-lapse movie

Yeast cultures were grown at 30°C in synthetic complete media with 2% glucose (SC) to OD600 0.2–0.6, and placed on 0.08 mm cover glass and covered with 1 mm thick 1.5% agarose (NuSieveTM GTGTM Agarose, Lonza) pad made with SC. Images were acquired using Zeiss Observer Z1 microscope with Axiocam 506 mono camera (Zeiss), ×63/1.4NA oil immersion objective and Tempcontrol 37-2 digital (PeCon) to keep the sample at 30°C. Time-lapse imaging was performed using ZEN software, an automated stage and Definite Focus. Phase-contrast, Citrine, and mCherry images were taken every 3 min for 8 hr from up to 12 positions. Colibri LED modules at 25% power were used to

excite the fluorescent proteins. Cdc6-Citrine was imaged using Zeiss Filter Set 46 and Colibri 505 LED module for 500 ms. Whi5-mCherry was imaged using Zeiss Filter Set 61 HE and Colibri 540–580 LED module for 750 ms. The analysis was carried out using MATLAB (MATLAB, RRID:SCR_001622). The cells were segmented and tracked based on the phase-contrast image, followed by quantification of the nuclear fluorescence signals as described in *Doncic et al., 2013*. The nuclear signal was obtained by applying a 2D Gaussian fit to the brightest point in the cell and assigning the 25% with highest signal as the nucleus. The cytoplasmic fluorescence signal was considered as background and was subtracted from the nuclear signal. For every strain, data are from at least two repeats with different transformants.

## Protein purification for Mcm2–7 loading assay

Sic1 was purified as described before (*Gros et al., 2014*). ORC and Cdt1·Mcm2–7 were purified as described previously (*Remus et al., 2009*). For Cdc6·Clb2·Cdk1·Cks1, cells (strain YDR12) were grown in 10 l of YPD at 25°C to a density of $2 \times 10^7$ ml$^{-1}$, arrested in G1 phase by addition of 50 µg ml$^{-1}$ α-factor, harvested by centrifugation, resuspended in 10 L YP/2% galactose lacking α-factor, and incubated at 25°C overnight to induce expression of Cdc6-TAP$^{TCP}$. Next day, cells were harvested by centrifugation, washed twice with cold 25 mM HEPES, pH 7.6/1 M sorbitol, once with buffer H (25 mM HEPES–KOH pH 7.6/0.02% NP-40/10% glycerol)/300 mM KCl, resuspended in 0.5× volume of buffer H/300 mM KCl/2 mM DTT/protease inhibitor cocktail, and frozen as droplets in liquid nitrogen. The resulting 'popcorn' was stored at −80°C until further processing. Cell lysate was prepared by crushing the frozen popcorn in a freezer mill (SPEX CertiPrep 6,850 Freezer/Mill) for 6 cycles of 2 min at a rate of 15 impacts per second. Crushed cell powder was thawed on ice, resuspended with 1 volume of buffer H/0.3 M KCl/1 mM DTT. Insoluble material was pelleted by centrifugation of the lysate in a type 45 Ti rotor (Beckman) for 60 min at 40,000 rpm. The clarified extract was supplemented with 2 mM CaCl$_2$ and incubated with 1 ml calmodulin affinity beads for 2 hr at 4°C. The calmodulin resin was washed with 10 CV of buffer H/300 mM KCl/2 mM CaCl$_2$/1 mM DTT, and bound protein eluted with 10 CV of buffer H/300 mM KCl/1 mM EDTA/2 mM EGTA/1 mM DTT. Peak fractions were pooled, concentrated by centrifugation through an Amicon spin concentrator, digested for 4 hr at 4°C with TEV protease to remove the TAP$^{TCP}$ tag, and subsequently fractionated by gel filtration using a 24 ml Superdex 200 column equilibrated in buffer H/300 mM KCl/1 mM EDTA/1 mM EGTA/1 mM DTT. Peak fractions were pooled, diluted with 2× volumes of buffer H/1 mM EDTA/1 mM EGTA/1 mM DTT to a final salt concentration of 100 mM KCl, and fractionated on a 1 ml Mono S ion exchange column using an elution gradient of 0.1–1 M KCl over 10 CV. Peak fractions were pooled and stored in aliquots at −80°C after snap freezing in liquid nitrogen. For the original identification of the Cdc6-associated proteins by mass spectrometry, purified complex was fractionated by SDS–PAGE, individual Coomassie-stained bands excised from the gel, and proteins subjected to mass-spectrometric analysis after in-gel trypsin digestion using standard protocols.

## Gel-filtration analysis of Sic interaction with Cdc6·Clb2·Cdk1·Cks1

6 µg of Sic1, 4 µg Cdc6·Clb2·Cdk1·Cks1, or a mixture of 6 µg Sic1 and 4 µg Cdc6·Clb2·Cdk1·Cks1 were incubated in a 50 µl reaction volume containing buffer H/300 mM KCl/1 mM EDTA/1 mM EGTA/1 mM DTT for 10 min at room temperature, fractionated on a 2.4 ml Superdex 200 PC 3.2/30 column, and fractions analyzed by SDS–PAGE and silver stain.

## Mcm2–7 loading assay

Purified Mcm2–7 loading onto linear 1 kbp ARS-containing DNA immobilized on paramagnetic beads using was performed as described previously (*Remus et al., 2009*), except that 50 nM Cdc6·Clb2·Cdk1·Cks1 was used in place of Cdc6. Where indicated, Sic1 was included at 500 nM.

## Prediction of IDRsin Cdc6 and Clb2

The IDRs and disordered binding regions of Cdc6 were predicted using DISOPRED (UCL Bioinformatics Group, RRID:SCR_010248; *Ward et al., 2004*), IUPRED2A (IUPRED, RRID:SCR_014632; *Mészáros et al., 2018*), PrDOS (PrDOS, RRID:SCR_021886), MFDp2 (MFDp2, RRID:SCR_021885; *Mizianty et al., 2013*), and CSpritz (CSpritz, RRID:SCR_021884; *Walsh et al., 2011*). A consensus was derived by combining the high confidence predictions from all these programs.

## Structure modeling of Cdc6 and Clb2

The atomic coordinates for Cdc6 isolated from the cryo-EM structure of ORC-Cdc6-Cdt1-Mcm2–7 intermediate (OCCM; PBD ID: 5V8F; *Yuan et al., 2017*) were extracted and used to build a full-length template of Cdc6. The 140 missing residues from this structure for Cdc6 was modeled separately via an ab initio modeling program QUARK (QUARK, RRID:SCR_018777; *Zhang et al., 2016*). The full-length Cdc6 model was constructed by combining abovementioned fragment and extracted coordinates of Cdc6 using MODELLER (MODELLER, RRID:SCR_008395; *Eswar et al., 2006*) and further refined using 3DRefine (*Bhattacharya et al., 2016*). Clb2 was modeled using a hybrid threading approach, ab initio, and template-based approach based on the algorithm, I-TASSER (Iterative Threading ASSEmbly Refinement) (I-TASSER, RRID:SCR_014627; *Roy et al., 2010*). I-TASSER identifies structural templates from the PDB by a multithreading server, LOMETS (LOMETS, RRID:SCR_021882; *Wu and Zhang, 2007*), and then constructs full-length atomic models by iterative template-based fragment assembly simulations. The IDR in Clb2 was modeled independently using QUARK (*Zhang et al., 2016*) and incorporated to create a full-length model in similar manas described above for Cdc6. Both the full-length modeled structures were meticulously evaluated by various structure verification programs, including VERIFY3D (University of California at Los Angeles – Department of Energy Institute for Genomics and Proteomics, RRID:SCR_001921; *Eisenberg et al., 1997*), Voromqa (VoroMQA, RRID:SCR_021881; *Olechnovič and Venclovas, 2017*), Proq3 (ProQ3, RRID:SCR_021880; *Uziela et al., 2016*), and ProsaWeb (ProSA-web, RRID:SCR_021879; *Wiederstein and Sippl, 2007*). These programs employ a variety of methods inclusive of deep learning approaches to evaluate the quality of the models. The final models were chosen by selecting the best evaluation profiles and their correlation with known functional features. The models were visualized and analyzed for their biophysical properties using the visualization programs Pymol (PyMOL, RRID:SCR_000305; *Schrodinger, 2010*) and ChimeraX (UCSF ChimeraX, RRID:SCR_015872; *Pettersen et al., 2021*).

## Acknowledgements

We would like to thank Lea Schroeder and Yooko Caroll for technical assistance, and Dr. Frederick Cross for anti-Clb2 antibody. This work was supported by PSC-CUNY enhanced award to AI, NIGMS grants R01-GM127428 and R01-GM107239 and NIH/NCI Cancer Center Support Grant P30 CA008748 to DR, ERC Consolidator Grant 649124, Centre of Excellence for 'Molecular Cell Technologies' TK143, and Estonian Science Agency grant PRG550 to ML, Francis Crick Institute, which receives its core funding from Cancer Research UK, the UK Medical Research Council, and the Wellcome Trust (FC001066) to JD.

## Additional information

### Funding

| Funder | Grant reference number | Author |
| --- | --- | --- |
| PSC-CUNY Enhanced Award | 64657-00 52 | Amy E Ikui |
| National Institute of General Medical Sciences | R01-GM127428 | Dirk Remus |
| National Institute of General Medical Sciences | R01-GM107239 | Dirk Remus |
| National Institutes of Health | P30 CA008748 | Dirk Remus |
| European Research Council | 649124 | Mart Loog |
| Centre of Excellence for 'Molecular Cell Technologies' | TK143 | Mart Loog |
| Estonian Research Council | PRG550 | Mart Loog |

| Funder | Grant reference number | Author |
|--------|------------------------|--------|
| Francis Crick Institute | FC001066 | John FX Diffley |

The funders had no role in study design, data collection, and interpretation, or the decision to submit the work for publication.

## Author contributions

Jasmin Philip, Conceptualization, Data curation, Formal analysis, Investigation, Methodology, Validation, Visualization, Writing - original draft, Writing – review and editing; Mihkel Örd, Data curation, Formal analysis, Investigation, Methodology, Validation, Writing - original draft, Writing – review and editing; Andriele Silva, Data curation, Investigation, Methodology, Validation, Writing - original draft; Shaneen Singh, Resources, Software, Supervision, Validation, Writing - original draft; John FX Diffley, Conceptualization, Funding acquisition, Project administration, Resources, Supervision, Validation, Writing – review and editing; Dirk Remus, Conceptualization, Data curation, Formal analysis, Funding acquisition, Investigation, Methodology, Project administration, Resources, Validation, Writing - original draft, Writing – review and editing; Mart Loog, Funding acquisition, Project administration, Resources, Supervision, Validation, Writing – review and editing; Amy E Ikui, Conceptualization, Data curation, Formal analysis, Funding acquisition, Investigation, Methodology, Project administration, Resources, Software, Supervision, Validation, Visualization, Writing - original draft, Writing – review and editing

## Author ORCIDs

Jasmin Philip ![ORCID] http://orcid.org/0000-0001-9030-9228
John FX Diffley ![ORCID] http://orcid.org/0000-0001-5184-7680
Dirk Remus ![ORCID] http://orcid.org/0000-0002-5155-181X
Amy E Ikui ![ORCID] http://orcid.org/0000-0002-2712-5351

## Decision letter and Author response

Decision letter https://doi.org/10.7554/eLife.74437.sa1
Author response https://doi.org/10.7554/eLife.74437.sa2

# Additional files

## Supplementary files

• Transparent reporting form
• Supplementary file 1. Strain list.

## Data availability

Original western blot gel images and quantification data have been deposited in Dryad under https://doi.org/10.5061/dryad.8931zcrrz.

The following dataset was generated:

| Author(s) | Year | Dataset title | Dataset URL | Database and Identifier |
|-----------|------|---------------|-------------|-------------------------|
| Ikui AE, Philip J, Ord M, Loog M | 2022 | Cdc6 is sequentially regulated by PP2A-Cdc55, Cdc14 and Sic1 for origin licensing in *S. cerevisiae* | http://dx.doi.org/10.5061/dryad.8931zcrrz | Dryad Digital Repository, 10.5061/dryad.8931zcrrz |

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
