## [Editor Report]

The work at focus in your manuscript shows that there are three mechanisms that coordinate the stability of Cdc6 to allow for the critical DNA replication licensing event that must occur before yeast cells enter the G1 phase and is an important contribution to the cell cycle field. How Cdc6 is dephosphorylated is complex and the reviewers all applaud the biochemical and molecular biology approaches taken. While no new experiments are required for the publication we call your attention to the three reviews below where recommendations for revisions are given.

---

## [Decision Letter]

**Decision letter after peer review:**

Congratulations, we are pleased to inform you that your article, "Cdc6 is sequentially regulated by PP2A-Cdc55, Cdc14 and Sic1 for origin licensing in *S. cerevisiae*", has been accepted for publication in *eLife*. Your article has been reviewed by 3 peer reviewers, and the evaluation has been overseen by a Reviewing Editor and Kevin Struhl as the Senior Editor. The following individual involved in review of your submission has agreed to reveal their identity: Martha Cyert (Reviewer #2).

*Reviewer #1:*

In this manuscript Ikui and colleagues investigate the mechanism of Cdc6 dephosphorylation during the cell cycle. Cdc6 is one of four proteins required to license eukaryotic origins of replication and is subject to tight regulation to ensure that this event is restricted to G1. More specifically, the CDK-dependent phosphorylation of two regions of Cdc6 (phospho-degrons) leads to the ubiquitin mediated degradation of Cdc6. Although the degradation of Cdc6 as cells go through the G1/S transition is straightforward to understand, the reaccumulation of Cdc6 at the M/G1 transition is less well understood. Here the authors demonstrate that at least three mechanisms coordinate to stabilize Cdc6 and prepare Cdc6 for helicase loading at the M/G1 transition: Clb2-Cdk1 binding, Cdc14 dephosohorylation, and Sic1 binding. The authors provide the strongest evidence for the first mechanism which they connect to the function of PP2A-Cdc55 phosphatase. The evidence for the role of Cdc14 and Sic1 is less extensive but nevertheless support the conclusions made. Finally, the authors propose a new structure of the Cdc6-Clb2 interaction based on bioinformatic analysis. This part of the paper is not well supported, lacking any experiments to test to proposed model. Overall, the manuscript will be an important contribution to our understanding of the M/G1 transition in yeast cells.

Specific points:

1. There is one particularly confusing aspect of the paper that the authors should resolve. Initially, the authors suggest that the N-terminal phospho-degron is shielded by Clb2-Cdk1 binding (Figure 2). However, if this were the case, then the Cdc6-T7 mutant should no longer be effected by either Clb2-Cdk1 binding or PP2A-Cdc55 function. Despite this expectation, the authors find that deletion of CDC55 results in increased Cdc6-T7 protein levels. Do the authors think that T7 can also act as a phospho-degron? Does this effect involve Clb2-Cdk1? It is important that these questions are addressed by the revised manuscript.

2. The evidence that the phosphospecific antibody recognizes the T7 site is strong but the test used does not show that it is specific for just that site (there are no other sites present). To show that it is specific for T7 the authors should show that the antibody does not recognize Cdc6 that is mutated at T7 and has all the other sites intact and modified. On a similar note, it is not clear that the very weak signal seen in the presence of Cdc55 is above background (Figure 3D, WT). A similar control showing that it goes away in the absence of the T7 phosphorylation site would make this more compelling.

3. The data that PP2A-Cdc55 and Cdc14 target different sites is very nice.

4. The data that Sic1 disrupts the Clb2-Ckd1-Cdc6 complex is clear. What is not shown is the involvement of Cks1 since Cks1 is not shown on any of the gels in D. Either the authors need to add experiments showing Cks1 (maybe a western blot) or limit their conclusions to the proteins that they see.

5. The last part of manuscript describing a potential structure of the Cdc6-Clb2 interaction is the weakest of the paper. Although the growing power of computational prediction is noteworthy, if the authors want to include this structure in the paper, it seems appropriate to perform a direct test of the proposed structure. For example, mutating regions predicted to mediate the interactions/Cdc6 degron protection and test their impact on the complex.

*Reviewer #2:*

In the paper by Philip et al., Cdc6 is sequentially regulated by PP2A-Cdc55, Cdc14 and Sic1 for origin licensing in *S. cerevisiae* the authors unravel distinct phosphatases (PP2A and CDC14) and show in detail how they regulate Cdc6 interaction with Clb2 (PP2A) and dephosphorylate a C-terminal degron (Cdc14) to stabilize Cdc14 in mitosis. Further the authors unravel a role for Sic1 the Cdk inhibitor in licensing, showing how it releases Clb2·Cdk1·Cks1 from Cdc6 which allows Mcm2-7 to load onto chromatin.

Major strengths of the paper are :

The clear determinations of Cdc6 protein levels both in WT cells and various mutants both by immunoblot and single cell analysis (Figure 1)

Use of phosphospecific antibody to Cdc6 T7 (Figure 3) to follow its level through the cell cycle and increase in mutants of Cdc55 (PP2A B subunit)

Beautiful in vitro dephosphorylatoin assays (Figure 4) with both PP2A-Cdc55 and Cdc14 to conclusively demonstrate the distinct sites in Cdc6 that they dephosphorylate

Beautiful in vitro chromatin binding assays (figure 5) to show role of Sic1 in dissociating Cdc6-Clb2-CDK to promote Mcm2-7 loading

Thorough analyses of Short linear motifs (SliMs) in Cdc6 and their roles in Protein protein interactions along with convincing structural modeling of Cdc6

In sum the authors unravel in glorious biochemical detail the logic by which Cdc6 Dephosphorylation contributes to its critical cell cycle regulation.

I can find no fault with this paper which was extremely well written, and congratulate the authors on a beautiful contribution that furthers our understanding of cell cycle roles for phosphatases. I wholeheartedly agree with their statement that 'Future research should focus on docking motifs for phosphatases and their regulatory subunits to understand phosphatase specificity" -not only in yeast but in all eukaryotes as phosphatase signaling pathways are much less well understood than their kinase cousins.

*Reviewer #3:*

In the study 'Cdc6 is sequentially regulated by PP2A-Cdc55, Cdc14 and Sic1 for origin licensing in *S. cerevisiae*', Philip et al. investigate the phosphorylation of the pre-replicative complex protein Cdc6. Cdc6 is phosphorylated and bound by Clb2-Cdk to prevent re-initiation of DNA replication. A detailed understanding of how Cdc6 is then dephosphorylated was missing. In the present manuscript, Philip et al. convincingly show that Cdc6 is dephosphorylated by both PP2ACdc55 and Cdc14, with PP2A specifically targeting the N-terminal sites, and Cdc14 the C-terminus. The authors present evidence that N-terminal dephosphorylation by PP2ACdc55 as well as inhibition by Sic1 leads to unbinding of Clb2, allowing Mcm2-7 loading.

Overall, this study gives valuable insights into the molecular mechanism responsible for Cdc6 dephosphorylation and the consequence on pre-RC assembly. The claims are convincingly supported by well-designed experiments.

1) The manuscript is well written. However, I feel it could benefit from a section in the discussion that puts the work and its implications into a more general context. While the discussion provides a lot of valuable detail, such an additional section would make the work better accessible to a broader audience.

2) In Figure 3D and E, the authors show a peak of Cdc6-T7 phosphorylation in cdc55-101 mutant cells that is absent in WT. This peak is mostly based on a single timepoint. Can we exclude the possibility that a similar peak in WT was simply missed due to slightly different cell-cycle timing? Also, the quantified data in Figure 3E seems to be based on a single experiment. I think replicates, ideally with higher time resolution, are needed.

3) The effect of cdc55 deletion on Cdc6-T7 phosphorylation shown in Figure 3C is surprisingly weak. Is this because most Cdc6-T7 is already phosphorylated in the WT? Can the fraction of phosphorylated Cdc6-T7 be estimated?

4) The authors need to provide more details on the microscopy experiments. What microscopy settings were used? How exactly were cells/nuclei segmented and the signal quantified?

5) The authors also need to provide more details on the flow cytometry experiments. What laser was used, where the data gated?

6) Based on their bioinformatics analysis, the authors propose a detailed mechanism for Cdc6-Clb2 interaction. Can this be tested experimentally?

7) The authors write: "Whi5 nuclear import was used as a cell cycle indicator that was set as time zero, making the Start point." I think that by 'Start' they refer to the first point of the time traces. However, this is confusing considering the cell cycle transition 'Start' is typically defined as the time of Whi5 export.

---

## [Author Response]

Reviewer #1:Specific points:1. There is one particularly confusing aspect of the paper that the authors should resolve. Initially, the authors suggest that the N-terminal phospho-degron is shielded by Clb2-Cdk1 binding (Figure 2). However, if this were the case, then the Cdc6-T7 mutant should no longer be effected by either Clb2-Cdk1 binding or PP2A-Cdc55 function. Despite this expectation, the authors find that deletion of CDC55 results in increased Cdc6-T7 protein levels. Do the authors think that T7 can also act as a phospho-degron? Does this effect involve Clb2-Cdk1? It is important that these questions are addressed by the revised manuscript.

It is true that Cdc6-T7 mutant whose phospho-degrons were mutated to alanine is affected by Cdc55 in Figure 2B. It suggests that there is another mechanism involved. We tested a possibility that Cdc6-T7 serves as a phosphor-degron (new results in Figure 3—figure supplement 1). Cdc6-T7 protein level was not altered in *cdc4-1* mutant, suggesting that T7 site might not be targeted by SCF complex. Interestingly, Ccd6-6A mutant was very unstable which was not rescued when Cdc4 was inactivated (Figure 3-Supplement 1). We conclude that Cdc6 has other degradation mechanisms than the known phospho-degrons independently of SCF^Cdc4^, which is now reflected in page 10-11 and Figure 3—figure supplemental 1.

2. The evidence that the phosphospecific antibody recognizes the T7 site is strong but the test used does not show that it is specific for just that site (there are no other sites present). To show that it is specific for T7 the authors should show that the antibody does not recognize Cdc6 that is mutated at T7 and has all the other sites intact and modified. On a similar note, it is not clear that the very weak signal seen in the presence of Cdc55 is above background (Figure 3D, WT). A similar control showing that it goes away in the absence of the T7 phosphorylation site would make this more compelling.

We agree that Cdc6-T7 signal in the wild type is background signal. We modified our text on page 11 to explain that Cdc6-T7 phosphorylation is background signal (Figure 3D). Thank you for the suggestion.

3. The data that PP2A-Cdc55 and Cdc14 target different sites is very nice.4. The data that Sic1 disrupts the Clb2-Ckd1-Cdc6 complex is clear. What is not shown is the involvement of Cks1 since Cks1 is not shown on any of the gels in D. Either the authors need to add experiments showing Cks1 (maybe a western blot) or limit their conclusions to the proteins that they see.

In page 14, we modified our conclusion that Cdc6 is released from the Clb2-Cdk1 complex instead of Clb2-Cdk1-Cks1 complex.

5. The last part of manuscript describing a potential structure of the Cdc6-Clb2 interaction is the weakest of the paper. Although the growing power of computational prediction is noteworthy, if the authors want to include this structure in the paper, it seems appropriate to perform a direct test of the proposed structure. For example, mutating regions predicted to mediate the interactions/Cdc6 degron protection and test their impact on the complex.

We mutagenized our Cdc6-E45 to alanine, Cdc6-E45A, using bioinformatic tools and showed that E45 enhances Cdc6-Clb2 intreaction. We found no Cdc6-Clb2 binding in the Cdc6-E45A mutant based on our bioinformatic analysis. However, it is still early stage of this mutagenesis analysis, therefore we would like to stay with the current Cdc6-Clb2 model in this manuscript.

Reviewer #2:In the paper by Philip et al., Cdc6 is sequentially regulated by PP2A-Cdc55, Cdc14 and Sic1 for origin licensing in *S. cerevisiae* the authors unravel distinct phosphatases (PP2A and CDC14) and show in detail how they regulate Cdc6 interaction with Clb2 (PP2A) and dephosphorylate a C-terminal degron (Cdc14) to stabilize Cdc14 in mitosis. Further the authors unravel a role for Sic1 the Cdk inhibitor in licensing, showing how it releases Clb2·Cdk1·Cks1 from Cdc6 which allows Mcm2-7 to load onto chromatin.

We thank the reviewer for the positive feedback.

Reviewer #3:In the study 'Cdc6 is sequentially regulated by PP2A-Cdc55, Cdc14 and Sic1 for origin licensing in *S. cerevisiae*', Philip et al. investigate the phosphorylation of the pre-replicative complex protein Cdc6. Cdc6 is phosphorylated and bound by Clb2-Cdk to prevent re-initiation of DNA replication. A detailed understanding of how Cdc6 is then dephosphorylated was missing. In the present manuscript, Philip et al. convincingly show that Cdc6 is dephosphorylated by both PP2ACdc55 and Cdc14, with PP2A specifically targeting the N-terminal sites, and Cdc14 the C-terminus. The authors present evidence that N-terminal dephosphorylation by PP2ACdc55 as well as inhibition by Sic1 leads to unbinding of Clb2, allowing Mcm2-7 loading.Overall, this study gives valuable insights into the molecular mechanism responsible for Cdc6 dephosphorylation and the consequence on pre-RC assembly. The claims are convincingly supported by well-designed experiments.1) The manuscript is well written. However, I feel it could benefit from a section in the discussion that puts the work and its implications into a more general context. While the discussion provides a lot of valuable detail, such an additional section would make the work better accessible to a broader audience.

We included a summary paragraph of the results that broadens the implications of the data we collected on page 17.

2) In Figure 3D and E, the authors show a peak of Cdc6-T7 phosphorylation in cdc55-101 mutant cells that is absent in WT. This peak is mostly based on a single timepoint. Can we exclude the possibility that a similar peak in WT was simply missed due to slightly different cell-cycle timing? Also, the quantified data in Figure 3E seems to be based on a single experiment. I think replicates, ideally with higher time resolution, are needed.

We rephrased the results based on Review 1’s comments that Cdc6-T7 phosphorylation is suppressed in wild type cells in page 11. We also added flow cytometry results (Figure 3—figure supplement 2) to support Figure 3D to show that both strains are in mitosis at 80 min timepoint.

3) The effect of cdc55 deletion on Cdc6-T7 phosphorylation shown in Figure 3C is surprisingly weak. Is this because most Cdc6-T7 is already phosphorylated in the WT? Can the fraction of phosphorylated Cdc6-T7 be estimated?

We think this is because cells have high Cdk1 activity due to nocodazole treatment to induce mitotic arrest in Figure 3C. T7 is already phosphorylated in *CDC55* cells during mitosis, which promotes Cdc6-Clb2 association.

4) The authors need to provide more details on the microscopy experiments. What microscopy settings were used? How exactly were cells/nuclei segmented and the signal quantified?

We included additional details for the microscopy experiment in the revised manuscript on page 26.

5) The authors also need to provide more details on the flow cytometry experiments. What laser was used, where the data gated?

We added the channel used in the revised manuscript on page 25. Our data are ungated.

6) Based on their bioinformatics analysis, the authors propose a detailed mechanism for Cdc6-Clb2 interaction. Can this be tested experimentally?

We do not have experimental system to test our hypothesis at this moment.

7) The authors write: "Whi5 nuclear import was used as a cell cycle indicator that was set as time zero, making the Start point." I think that by 'Start' they refer to the first point of the time traces. However, this is confusing considering the cell cycle transition 'Start' is typically defined as the time of Whi5 export.

We made this correction on page 7. It is now called “zero” point instead of “Start” to avoid the confusion.